



# Ensemble Streamflow Data Assimilation using WRF-Hydro and DART: Hurricane Florence Flooding

Mohamad El Gharamti[1], James L. McCreight[1], Seong Jin Noh[2], Timothy J. Hoar[1], Arezoo RafieeiNasab[1], and Benjamin K. Johnson[1]

[1]National Center for Atmospheric Research, Boulder, CO, USA
[2]Kumoh National Institute of Technology, Gumi, South Korea

**Correspondence:** Mohamad El Gharamti (gharamti@ucar.edu)

**Abstract.** Predicting major floods during extreme rainfall events remains an important challenge. Rapid changes in flows over short time-scales combined with multiple sources of model error makes it difficult to accurately simulate intense floods. This study presents a general data assimilation framework that aims to improve flood predictions in channel routing models. Hurricane Florence, which caused catastrophic flooding and damages in the Carolinas in September 2018, is used as a case study.

The National Water Model (NWM) configuration of the WRF-Hydro modeling framework is interfaced with the Data Assimilation Research Testbed (DART) to produce ensemble streamflow forecasts and analyses. Hourly streamflow observations from 107 United States Geological Survey (USGS) gauges are assimilated for a period of one month.

The data assimilation (DA) system developed in this paper explores two novel contributions: (1) Along-The-Stream (ATS) covariance localization and (2) spatially and temporally varying adaptive covariance inflation. ATS localization aims to mitigate

not only spurious correlations, due to limited ensemble size, but also physically incorrect correlations between unconnected and indirectly connected state variables in the river network. We demonstrate that ATS localization provides improved information propagation during the model update. Adaptive prior inflation is used to tackle errors in the prior, including large model biases. Analysis errors incurred during the update are addressed using posterior inflation. Results show that ATS localization is a crucial ingredient of our hydrologic DA system, providing at least 40% more accurate (RMSE) streamflow estimates than

regular, Euclidean distance-based localization. Assessment of hydrographs indicates that adaptive inflation is extremely useful and perhaps indispensable for improving the forecast skill during flooding events with significant model errors. We argue that adaptive prior inflation is able to serve as a vigorous bias correction scheme which varies both spatially and temporally. Major improvements over the model's severely underestimated streamflow estimates are suggested along Pee Dee River in South Carolina and many other locations in the domain, where inflation is able to avoid filter divergence and thereby assimilate

significantly more observations.

## 1    Introduction

Affecting nearly a hundred million people worldwide per year, flooding is the most common natural disaster (Guha-Sapir et al., 2013). Flooding impacts human life, livelihood, and property. Improved streamflow flood forecasts can benefit the public



in a variety of ways from planning to emergency management. The topic of flood forecasting remains an area of active of
research and operational development. This study contributes to improving short-term (hourly) streamflow flooding forecasts
by minimizing error in their initial conditions through streamflow data assimilation. We focus on rainfall-driven streamflow
flooding caused by Hurricane Florence in 2018. Within the context of an operational and spatially distributed hydrologic model,
we examine the DA challenges of dominant errors (bias) arising from the precipitation boundary conditions (forcings) and of
improving information propagation from the observations into the model ensemble (background).

Streamflow is one of the most commonly observed hydrologic variables. Its earliest measurements date back to the late
nineteenth century (Ashman et al., 2004). In more recent decades, the assimilation of streamflow observations into hydrologic
models has followed various DA approaches and covered the gamut of applications (e.g., Wood and Szöllösi-Nagy, 1978; Ki-
tanidis and Bras, 1980; Moradkhani et al., 2005; Weerts and El Serafy, 2006; Clark et al., 2008; Pauwels and De Lannoy, 2009;
Seo et al., 2009; DeChant and Moradkhani, 2012; Noh et al., 2013; McMillan et al., 2013; Rafieeinasab et al., 2014; Lee and
Seo, 2014; Sun et al., 2015; Ercolani and Castelli, 2017; Abbaszadeh et al., 2018; Ziliani et al., 2019, and references therein).
Meanwhile, hydrologic models have evolved from lumped to high-resolution and spatially distributed given the increase in
computational resources and the availability of high-resolution terrain and forcing data.

While research studies have shown success of streamflow and even multivariate data assimilation, operational flood fore-
casting systems do not typically employ data assimilation (Emerton et al., 2016). Liu et al. (2012) detail the hurdles between
hydrologic forecasting research and operations. The authors provide a wealth of recommendations on transitioning DA into
hydrologic forecasting. A primary reason DA research is not commonly applied in operational settings is that the methods can
perform poorly in the presence of large model errors. Probabilistic DA approaches are *optimal* only when the model is unbiased
(Dee, 2005). When large biases or systematic errors are present, the methods commonly result in "filter divergence," the case
when observations are unable to provide state updates to the model background (e.g., DeChant and Moradkhani, 2012). In ad-
vocating for the adoption of DA methods in operational hydrologic forecasting, Seo et al. (2009) and Liu et al. (2012) suggest
blending DA procedures with other kinds of interventions in order to balance the need for operational robustness against the
multiple advantages (skill improvements, reproducibility, etc) offered by automated DA methods.

Divergence and other filter-related issues are often pronounced in rainfall-driven flooding systems. This is because the large
errors arising at the model boundary are often not part of the prognostic state of the system, have no memory and cannot
be constrained during the analysis. To overcome this, multiple bias correction strategies have been pursued in the context of
DA for flood forecasting. Joint state-parameter estimation is often applied to help mitigate model errors (e.g., Abbaszadeh
et al., 2018). Multiplicative bias correction parameters, to adjust forcing errors, are sometimes estimated alongside the physical
state and parameters (e.g., Seo et al., 2003). Bias-aware Kalman filters are applied to estimate model and observation bias
by implementing a separate update for two moments, the mean and the bias (e.g., Drécourt et al., 2006; Rasmussen et al.,
2016; Ridler et al., 2018). In addition, a conditional bias-penalized Kalman filter was developed for improved estimation and
prediction of hydrologic extremes. The filter operates by minimizing a weighted sum of error variances and Type-II squared
errors, different from the conventional Kalman filter which is based on least square minimization (Seo et al., 2018; Lee et al.,
2019; Jozaghi et al., 2019). In a recent study, Emery et al. (2020a) proposed updating the boundary fluxes based on the





differences between observed and prior streamflow. The authors rerun the assimilation forecast step with the updated boundary
conditions to produce a second prior without involving the streamflow state in the update. In this study, we explore the use of
spatially and temporally varying adaptive covariance inflation (El Gharamti, 2018) as a way to mitigate bias, in the context
of extreme flood simulations. Inflation helps restore spread in the ensemble which can yield a better fit to the observations
during the analysis. In addition, spatially varying inflation can help enhance the rank of the sample background covariance
matrix (El Gharamti, 2018). In soil and groundwater hydrology, using inflation was reported successful by multiple studies
(e.g., Bauser et al., 2018; Jamal and Linker, 2020). In surface water hydrology, however, the impact of inflation on streamflow
predictions is not fully understood. We note that the approach of Emery et al. (2020a), a temporally-fixed inflation parameter
(scalar) as a means of tuning static background error covariances, is significantly different from the temporally and spatially
adaptive inflation applied to time evolving background error covariances in this paper.

It is no surprise that many hydrologic and flood forecasting DA studies have highlighted the importance of estimating ac-
curate background error covariances. Model bias, as discussed above, and sampling error hinders proper estimation of error
covariances. The nonlinear relationship between variables in hydrologic modeling makes it further challenging to update un-
observed state variables. Filtering approaches only consider the instantaneous error covariances. Smoothers, on the other hand,
can be applied to remedy this problem (e.g., Pauwels and De Lannoy, 2006; Li et al., 2013). Even when employing an en-
semble smoother for flood forecasting, Rakovec et al. (2015) concluded that the elimination of the strongly nonlinear relation
between soil moisture and discharge observations improved flood forecasts. Clark et al. (2008) commented that modeled er-
ror correlations were much larger than observed error correlations and that inadequacies in modelling the spatial variability
of hydrological processes hindered the transfer of observational information to ungauged basins. In this study we revisit the
spatial basis for information propagation of observations via error covariances. We investigate updating distributed hydrologic
states following an along-the-stream (ATS) covariance localization which confines state updating to directly connected (de-
fined below) hydrologically states. Information propagation to ungauged basins (e.g. Sivapalan et al., 2003) within our strategy
requires such basins to be upstream of observations.

The development of the data assimilation framework in this paper begins from NOAA's National Water Model (NWM)
configuration of the WRF-Hydro hydroloigcal framework (Gochis et al., 2020). The NWM is a spatially distributed hydrologic
model that produces operational forecasts and analyses of distributed hydrologic states, including streamflow, over the con-
tinental United States (more recently, also with separate implementations in various other regions). The operational products
are not evaluated in this study. However, the real-time NWM forcing fields from Hurricane Florence are run through a model
configuration very close to the operational analysis configuration, providing an open loop (no DA) deterministic analysis very
close to what the NWM would have produced in real-time (if such an open-loop run were operational). Using one-way fluxes
from this analysis, we drive a "channel+bucket" submodel of the NWM that includes streamflow and conceptual bucket storage
states. The reduced computational cost of this submodel, the perturbation of its parameters, and the time-varying perturbations
applied to the deterministic fluxes from the land surface model provide an ensemble basis for our DA experiments. The re-
sulting NWM channel+bucket modeling system is interfaced with the Data Assimilation Research Testbed (DART). DART,
developed and maintained at the (United States) National Center for Atmospheric Research, is an open-source community





facility that provides software tools for data assimilation research, development, and education (Anderson et al., 2009). Hourly
streamflow observations from USGS gauges, retrieved from the (United States) National Water Information System (NWIS),
are used to update the spatially distributed ensemble states of streamflow and groundwater bucket head. The resulting analyses
could serve as the initial conditions for short term flood forecasts. The analyses are evaluated in terms of the assimilation priors
(i.e., the one hour forecast). Hydrographs and other time-series assessment tools (including errors, bias, ensemble spread, etc)
are utilized to investigate the performance of the DA framework. Streamflow distribution in space resulting from DA is also
studied and compared to the model's estimate.

The rest of the paper is organized as follows. Section 2 presents the Hurricane Florence subdomain, the NWM submodel
and its components, the uncertainties incorporated into the ensemble design and the USGS observations assimilated. DART
is briefly introduced in Section 3 and then ATS localization and adaptive inflation are described in Sections 3.2 and 3.3,
respectively. Spatial assessment with particular focus on bias correction is given in Section 3.6. A summary of the findings and
further discussions are found in Section 4.

## 2  Hydrologic Model and Data

### 2.1  National Water Model Subdomain: Hurricane Florence

In this study, we focus on a regional subdomain of the NWM CONUS domain affected by Hurricane Florence in September,
2018. Fig.1 shows this domain located over the states of North and South Carolina. Hurricane Florence reached Saffir-Simpson
category 4 strength on two separate dates prior to landfall. Though it weakened to category 1 by the time it made landfall, it
wreaked over $20 Billion US Dollars of damage largely attributed to inland freshwater flooding resulting from extreme rainfall.
Many of the largest flood peaks occurred after the dissipation of the hurricane on September 19, as water concentrated along
its course to the sea. A time line of observed and modeled attributes is shown in Table 1.

| Observed and Modeled Timeline | | |
|---|---|---|
| Modeled | NLDAS2-based restart, advance with operational forcings | August 1 |
| **Observed** | Hurricane forms at sea | August 31 |
| Modeled | Hourly data assimilation starts | September 1 |
| **Observed** | Hurricane landfall | September 14 |
| **Observed** | Hurricane dissipates | September 19 |
| Modeled | End of simulation | October 15 |

**Table 1.** Hurricane Florence timeline, Carolinas, USA, 2018. NLDAS2 denotes the forcing data for phase 2 of the North American Land
Data Assimilation System.







**Figure 1.** Florence model domain showing the river network with locations of the main Cape Fear and Neuse Rivers in North Carolina. The assimilated 107 gauges, from USGS, are denoted by grey dots. Twelve gauges displayed with red markers are used for diagnostic and validation purposes. The borders between the states of Virginia, North Carolina and South Carolina are also shown in black. The thickness of the river reaches denote the strength of streamflow (resulting from an open loop run and averaging over the month of September, 2018) such that larger thickness means higher streamflow.

Fig.1 shows the roughly 67,000 reaches of the NWM stream channel network in this subdomain. The NWM channel network
is based largely on USGS's National Hydrography Dataset; namely NHDPlus version 2 (McKay et al., 2012). The rectangular extent of the subdomain indicates the region over which atmospheric forcing data are used to drive the Noah-MP (Niu et al., 2011) land surface model (1km) and its two-way coupling to lateral surface and subsurface flow routing schemes (250m; Gochis and Chen, 2003) used by the NWM Analysis and Short-Range Forecast configurations (https://water.noaa.gov/about/nwm). As shown in Fig. 2, the lateral flow components are one-way coupled to the streamflow, reservoir and bucket models of the NWM.
Detailed description of this submodel can be found in following section.



All model code, domain data and parameter sets used in this study correspond to NWM version 2.0. The single exception is the groundwater bucket model formulation and parameters which are based on NWM version 2.1. We run the equivalent of the NWM "standard analysis and assimilation" cycle without the streamflow nudging used by the NWM. This is the configuration also used for the short and medium range forecast cycles. During the time of the study, the NWM extended analysis cycle had
not yet been implemented.

The major rivers in this region are labeled in Fig. 1. To the west, the Pee Dee river has its headwaters between Charlotte and Winston-Salem and flows from North Carolina into South Carolina. Its major tributary, the Lumber River, meets Pee Dee just before reaching the edge of the domain. The Cape Fear River, with its headwaters near Greensboro and joining the sea near Wilmington, is seen in the center of the domain. Further to the East, Neuse river flows by Durham and Raleigh (the capital of
North Carolina) and flows in to the Pamlico Sound. The Tar River (not labeled except by its cluster of 3 gauges in the legend) lies to the North of the Neuse, also flowing into the Pamlico Sound.

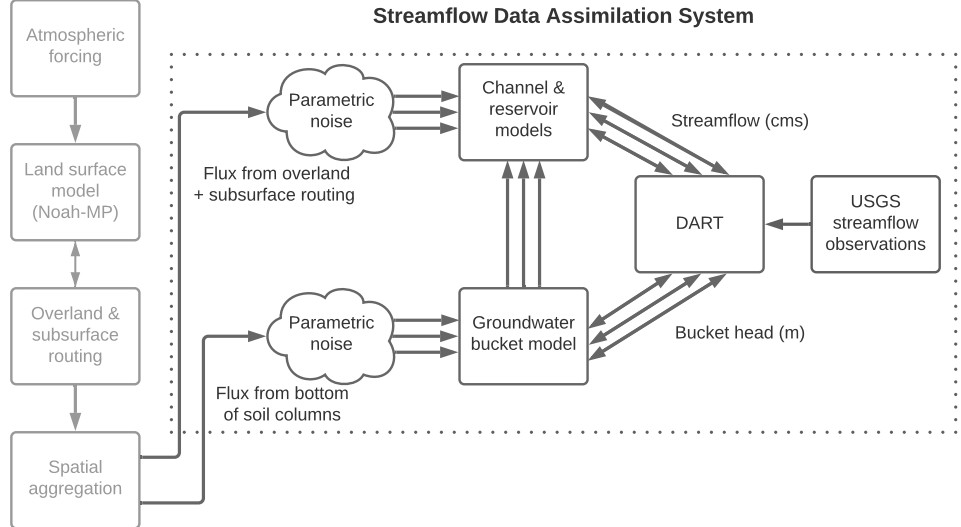

**Figure 2.** Streamflow data assimilation system overview. Vertical boxes on the left show the deterministic NWM model chain from forcing through aggregation of (overland, subsurface, and column drainage) routing output fluxes. These output fluxes are the inputs to the data assimilation system used in this paper, shown inside the dotted box. Random noise is applied to these inputs to generate ensemble forcings. Ensembles are denoted by groups of three arrows (the ensemble size is much larger than 3). The ensemble fluxes drive the ensemble model components (channel and reservoir model and groundwater bucket model) used in the assimilation. The groundwater states produce additional fluxes to the channel and reservoir model. The spatially distributed streamflow and bucket head states comprise the "state" vector passed to DART for updating by USGS streamflow observations.



## 2.2 Channel + Bucket Submodel

We run the so-called channel+bucket submodel of the NWM. Fig. 2 illustrates the one-way runoff fluxes to the streamflow and groundwater bucket models from the "upstream" model components (land surface model, overland and subsurface routing, and spatial aggregations). The fluxes are saved as forcing files for running the channel+bucket submodel used in our assimilation approach. To generate these fluxes, the full NWM is first run once with its own set of atmospheric boundary conditions.

The use of the one-way coupled submodel means that no error covariances with the "upstream" components of the model will be considered (e.g. soil moisture, surface head, etc) and that the control vector will consist of two spatially distributed states, streamflow and groundwater bucket head. Reservoir states, embedded in the stream network calculations, are not considered in the state updating.

## 2.3 Forcings, Spinup and Simulation

The full model was run (with no data assimilation) using NLDAS2 (Xia et al., 2014) forcings from 2010-10-01 to 2018-07-01. This open loop run was then continued from 2018-07-01 through 2018-10-15 with the NWM operational analysis and assimilation cycle forcings which were collected in real-time from NOMADS (NOAA Operational Model Archive and Distribution System). This real-time forcing product is based on MRMS (Zhang et al., 2016) gauge-adjusted and radar-only observed precipitation products along with short-range RAP and HRRR products (Benjamin et al., 2016, see https://water.noaa.gov/about/nwm). For the period 2018-08-01 through 2018-10-15, fluxes were saved for forcing the channel+bucket submodel in the data assimilation experiments. Fig. 2 shows these fluxes as inputs to the data assimilation system. Initial states for the data assimilation experiments were also taken from the full model run on 2018-08-01.

## 2.4 Muskingum-Cunge Streamflow Model

The NWM implements Muskingum-Cunge (M-C) streamflow routing with variable parameters (e.g., Ponce and Yevjevich, 1978) in a compound channel (Garbrecht and Brunner, 1991). M-C with variable parameters is a common approach to streamflow routing over large watersheds and has been successfully applied in many instances. The compound channel (Fig. 3) provides a lower trapezoidal channel and an upper rectangular channel section to simulate overbank flows. M-C is applied to the stream channel network derived from NHDPlus version 2, shown in Fig. 1, with trapezoidal channel geometry and Manning's N (roughness) parameter values for each reach.

The one-dimensional storage ($S$) relationship between inflow ($I$) and outflow ($O$) on a reach (spatial segment) is given by:

$$S = K\left[XI + (1-X)O\right], \tag{1}$$

with storage coefficient $K$ and weighting factor $X$. Formulated as a finite difference over a reach, this yields an explicit solution:

$$O_k = C_1 I_{k-1} + C_2 I_k + C_3 O_{k-1} + C_4 L, \tag{2}$$





for the current out flow ($O_k$, $k$ denoting the time step index) as a function of previous and current inflows ($I_{k-1}$ and $I_k$, respectively), previous outflow ($O_{k-1}$), and the lateral inflows (combined overland and subsurface, $L$) to the reach. The coefficients in equation 2 can be found in the literature, expressed as combinations of the respective Courant and Reynold's numbers:

$$C = \frac{c\Delta t}{\Delta x} = \frac{\Delta t}{K}, \tag{3}$$


$$D = \frac{q}{sc\Delta x} = 1 - 2X, \tag{4}$$

where $\Delta t$ is the time step, $\Delta x$ is the reach length, $s$ is the reach slope, $c$ is the celerity and $q$ is the unit discharge (discharge divided by the top width of the flow). Also shown are the relationships to $K$ and $X$ parameters.

The assumptions of M-C approach do not allow for backwater effects in the solution. However, the M-C variable parameter approach allows nonlinear flood wave dynamics by accounting for the interdependence of the time-varying flow rate and its geometry. Specifically, the celerity and unit discharge


$$c = \frac{dQ}{dA}, \tag{5}$$

$$q = \frac{Q}{b}, \tag{6}$$

used for calculating the coefficients in Equation 2, depend on the flow ($Q$) and its area ($A$) and top width ($b$) which are mediated by the channel geometry and roughness parameters in each reach. These parameters, some of which are shown in Fig. 3, are described in more detail in Section 2.6. The equation for celerity can be solved from Manning's equation for uniform flow. Garbrecht and Brunner (1991) solve the celerity equation for the case of the compound channel shown in Fig. 3. The variable parameter approach is an iterative solution, updating the flow and its geometry in alternate steps, to converge on a physically consistent discharge-geometry solution. The implementation in the NWM follows the "secant method" which takes a high and a low departure from an initial water depth and iterates through the equations of geometry and flow until the calculated flow rates converge within some threshold. Before the flow rates converge, their differences are used to reduce the discrepancy in the estimated water depths.


We note that the NWM makes a "short timestep approximation," $I_k = I_{k-1}$, to eliminate the spatial/topological dependence at the current time and render the solution of the M-C method embarrassingly parallel.

Reservoir objects embedded into the NWM routing network accept fluxes from the streamflow network and from the overland and subsurface routing model on adjacent grid cells. Water is discharged to the stream network via equations for both weir and orifice flow in the NWM "level pool" scheme. Because we do not include the reservoir level in the assimilation state vector, the reader is referred to Gochis et al. (2020) for further details.


## 2.5 Groundwater Bucket Model

Even when lateral routing processes are included in hydrologic modeling, deficiencies in soil and aquifer data and model process representations commonly lead to underestimation of the baseflow component of streamflow. The NWM employs a groundwater bucket model as a simple aquifer representation to mitigate this baseflow problem. This model accepts water






fluxes from the bottom of the land model's soil columns. The spatial representation of the buckets is derived from the NHDPlus
(McKay et al., 2012) catchments. These map roughly on to the stream reaches (with certain exceptions). The buckets have an
average areal extent of $\sim 3$ km across the NWM CONUS domain and therefore accept fractions of discharge from multiple land
model columns. The mapping from the land surface model to the buckets is performed by "user defined mappings" capability
of the model.

The bucket scheme is simple and highly conceptual. For this reason, calibration of its parameters is critical for reasonable
model simulations. The groundwater bucket model and its parameters are expressed by the following set of equations, which
are the only model components taken from NWM v2.1 (instead of v2.0). The current bucket head, $z_k$, is solved from the
previous bucket head, $z_{k-1}$, plus the change in head due to the bucket inflow $\widetilde{I}_k$:

$$z_k = z_{k-1} + \frac{\widetilde{I}_k \Delta t}{\widetilde{A}}, \tag{7}$$

where $\widetilde{A}$ is the bucket area. The finite capacity of the bucket is expressed in terms of a maximum head, $z_{\max}$, a tunable
parameter. When the current head exceeds this threshold, the $Q_{\mathrm{spill}}$ term becomes non-zero, discharging all excess head in a
single timestep.

**if** $z_k <= z_{\max}$ **then**

$\quad Q_{\mathrm{spill}} = 0$

**else**

$\quad z_{\mathrm{spill}} = z_k - z_{\max}$ \hfill (8)

$\quad z_k = z_{\max}$ \hfill (9)

$\quad Q_{\mathrm{spill}} = \dfrac{\widetilde{A} z_{\mathrm{spill}}}{\Delta t}$ \hfill (10)

**end if**

Head up to and including the $z_{\max}$ results in bucket discharge following an exponential equation, containing two additional
tunable parameters, $E$ (unitless) and $G$ (m$^3$/s):

$$Q_{\exp} = G\left[\exp\left(E\frac{z_k}{z_{\max}}\right) - 1\right]. \tag{11}$$

The spill discharge and the exponential bucket discharge are finally combined to give the total bucket outflow at the current
time step ($\widetilde{O_k}$) and the depth of water corresponding to $Q_{\exp}$ is removed from the bucket.

$$\widetilde{O_k} = Q_{\mathrm{spill}} + Q_{\exp}, \tag{12}$$
$$z_{k+1} = z_k - \frac{Q_{\exp}\Delta t}{\widetilde{A}} \tag{13}$$

Calibration of the bucket parameters (in advance of NWM version 2.1 calibration) yielded the following spatially uniform
bucket parameters used in this study: $C = 0.005$, $E = 7.1244$, and $z_{\max} = 15.6476$.



## 2.6 Sources of Uncertainty: Ensemble Design

We construct an ensemble of 80 members. This number was selected to balance computational demands and statistical performance. We did not investigate the effect of ensemble size on the results within this study. Incorporating different sources
of uncertainty into the ensemble is necessary to create variability and to obtain a good estimate of the background error covariance. Background error covariances are considered amongst and between the spatially distributed streamflow and bucket states ($O_k$ and $z_k$, respectively). We produce error distributions in these states through *a priori* error distributions on: 1) stream channel parameters, 2) forcing fluxes to the channel reaches and 3) forcing fluxes to the buckets.

The error distribution imposed on the streamflow channel parameters is time invariant and unaffected by the state update.
This kind of error source is termed as "multiphysics" (Berner et al., 2011), meaning that each member runs a different physical configuration of the model. The parameters shown in Fig. 3 describe the compound channel geometry used in the NWM v2.0: top width ($T$), bottom width ($B$), side slope ($m$), Manning's N ($n$), width of the compound channel ($T_{cc}$), and Manning's N of the compound channel ($n_{cc}$). While the lower part of the compound channel is trapezoidal, the upper part of the channel is assumed to be rectangular and therefore has no side slope parameter. These parameters vary in space and we define a scalar
multiplier for each parameter and ensemble member to generate a perturbed parameter vector from the existing NWM parameter vector. The multipliers are sampled from uniform distributions and in the case of 3 parameters, we redraw the multiplier until the following physical constraints are satisfied: $n_{cc} > 1.5n$, $T > 1.2B$ and $T_{cc} > 2T$. For the geometric quantities we draw multipliers from $\mathcal{U}[0.6, 1.4]$. For the Manning's N parameters we draw multipliers from $\mathcal{U}[0.8, 1.8]$ based on the prior belief that the original values are somewhat too low.

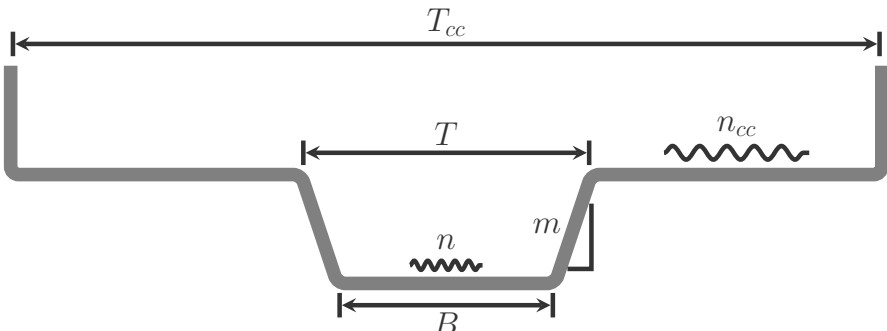

**Figure 3.** Schematic of the geometry and roughness parameters of the streamflow compound channel: top width ($T$), bottom width ($B$), side slope ($z$), Manning's N ($n$), width of the compound channel ($T_{cc}$) and Manning's N of the compound channel ($n_{cc}$).

Perturbations of the boundary fluxes to the streamflow and bucket models are applied at the hourly forcing time step. These perturbations are uncorrelated (in space, time, and member) Gaussian samples with zero mean and standard deviation equal to 40% of the flux value at each location. When the perturbations are added to the fluxes, a minimum of zero flux is ensured. Random noise generators are seeded as a function of datetime and ensemble member to ensure identical forcing distributions





are used across all experiments. Finally, perturbations are applied to the model initial states on September 1, 2018. However,
these ensemble initial conditions account for very little of the uncertainty in the overall experiment.

## 2.7 USGS Streamflow Observations

The USGS streamflow observations used by the NWM are provided along with its output in near-real-time on NOMADS.
The streamflow observations in these files, which correspond very closely to the values assimilated by the NWM, are always
"provisional" because they are near-real-time and they are subject to revision until they have been thoroughly assessed. For
this study, we collected NWM observation files as well as revised values from the USGS's NWIS many months after the time
period of this study. As expected, there were significant revisions to the streamflow values in the months following Hurricane
Florence. While it is important to study the differences between the provisional and approved data, here we chose to focus
exclusively on the revised observations. However, we note that the difference between these observation sets had a significant
impact on our results.

Fig. 1 shows the 107 USGS streamflow gauges used for evaluation in this study in green and red. The names of the gauges in
red are given in the caption as these locations are specifically called out in the results. All stream gauges considered in this study
have their contributing area entirely contained within this subdomain. All experiments presented here use the heteroskedastic
error model of 20% of the observed flow for the observation error ($m^3$/s). This is certainly a simplistic approach, but the
magnitude is roughly in line with previous studies (e.g. Coxon et al., 2015). We note that while important, the observation error
plays a somewhat secondary role in the quality of the assimilation, particularly given the application of inflation.

## 3 Data Assimilation Framework and Results

### 3.1 DART

This study uses the Data Assimilation Research Testbed (DART, Anderson et al., 2009) to perform ensemble Kalman filtering
for streamflow forecasting. Utilizing Bayes' rule, the goal is to sequentially use streamflow gauge data to guide the trajectory of
the hydrologic model towards a better flood prediction. The procedure consists of successive forecast and analysis steps. During
the forecast, a set of model realizations of the state variables are integrated forward in time using the nonlinear hydrologic
model:

$$\mathbf{x}_k^{f(i)} = \mathcal{M}\left(\mathbf{x}_{k-1}^{a(i)}, \boldsymbol{\theta}^{(i)}, \boldsymbol{\gamma}_k^{(i)}\right), \quad i = 1, 2, \ldots, N_e \tag{14}$$

where $\mathbf{x}_k = [\mathbf{O}_k, \mathbf{z}_k]^T$ is the DART state consisting of the streamflow and the bucket distributions. The superscript $i$ is the
number of the ensemble member, $N_e$ is the ensemble size, $f$ denotes forecast (prior) and $a$ is the analysis (posterior). The func-
tion $\mathcal{M}$ refers to the WRF-Hydro submodel. $\boldsymbol{\theta}$ and $\boldsymbol{\gamma}$ denote a set of physical parameters and input model forcings (described
in Section 2.6), respectively. As the data become available, DART assimilates the observations serially and applies an EAKF





update (Ensemble Adjustment Kalman Filter, Anderson, 2003) as follows:

$$\Delta\mathbf{x}_j^{(i)} = \sigma_{xy}\sigma_y^{-2}\Delta y^{(i)}, \quad j=1,2,\ldots,N_x \tag{15}$$

$$\mathbf{x}_{j,k}^{a(i)} = \mathbf{x}_{j,k}^{f(i)} + \alpha\Delta\mathbf{x}_j^{(i)}. \tag{16}$$

The observation increments, $\Delta y^{(i)}$, are first computed and then used to obtain the increments to the state, $\Delta x_j^{(i)}$. $\sigma_{xy}$ denotes the prior covariance of the observed variable, $y$, and the $j^{\text{th}}$ element in the state vector, $\mathbf{x}$. The total number of elements in the state is denoted by $N_x$. The sample variance of the observed variable is $\sigma_y^2$. A localization coefficient, $0 \leq \alpha \leq 1$ is used to limit the impact of spurious correlations in the update. $\alpha$ is computed as a function of the distance between observation and state variables given a predefined correlation structure.

Streamflow gauges are available at the location of the state variables, and assumed representative of the model element to which they are associated. This makes the (forward) observation operator linear and equal to the identity matrix, significantly simplifying the implementation of the update step in DART. Variance underestimation is tackled through covariance inflation such that the ensemble right after the forecast or analysis steps is inflated around its mean:

$$\mathbf{x}_j^{f|a(i)} = \sqrt{\lambda}\left(\mathbf{x}_j^{f|a(i)} - \overline{\mathbf{x}}_j^{f|a}\right) + \overline{\mathbf{x}}_j^{f|a}, \tag{17}$$

where $\overline{\mathbf{x}}_j$ is the $j^{\text{th}}$ element of the ensemble mean and the notation $f|a$ is used to refer to either forecast or analysis ensemble. The inflation factor $\sqrt{\lambda}$ (typically larger than 1) yields a sample covariance matrix scaled by $\lambda$.

## 3.2 Along-The-Stream Localization

It is well-recognized that the use of small ensemble sizes produces imperfect sample covariance matrices (e.g., Houtekamer and Mitchell, 2001). In fact, with a small ensemble the probability density function of the state remains only partially explored which can possibly yield loss of information and even filter divergence. In addition, the sample covariance would generally be contaminated with spurious unrealistic correlations that may degrade the quality of the Kalman update.

To overcome these issues, we resort to using covariance localization. The idea is to taper any spurious correlations between variables that are physically far from each other and are possibly uncorrelated, using $\alpha$ in equation (16). Studies have shown that given the Euclidean distance between different variables, a correlation function could be utilized to compute a localization factor, $\alpha$. In the present study, a simple Euclidean distance could be inappropriate in many circumstances. For example, reaches from two different watersheds could be physically close but be highly unrelated, particularly in terms of their error correlations. To this end, a topologically based localization strategy that adheres to the river network structure is applied. We introduce the Along-The-Stream Localization (ATS Localization) strategy. The idea is that only the reaches upstream and downstream from an particular observation are considered during the update (Fig. 4). The localization factor, $\alpha$, is computed using a selected functional form (e.g. Gaspari-Cohn, boxcar, or ramped-boxcar, see inset Fig. 6) which depends on the distance between any two reaches and the tunable localization radius, $r$.

ATS localization highlights some key features: (i) Downstream from each observation, we assume that the flow of information only travels downstream and not also back upstream. As such, we obtain tree-like shapes where the number of close



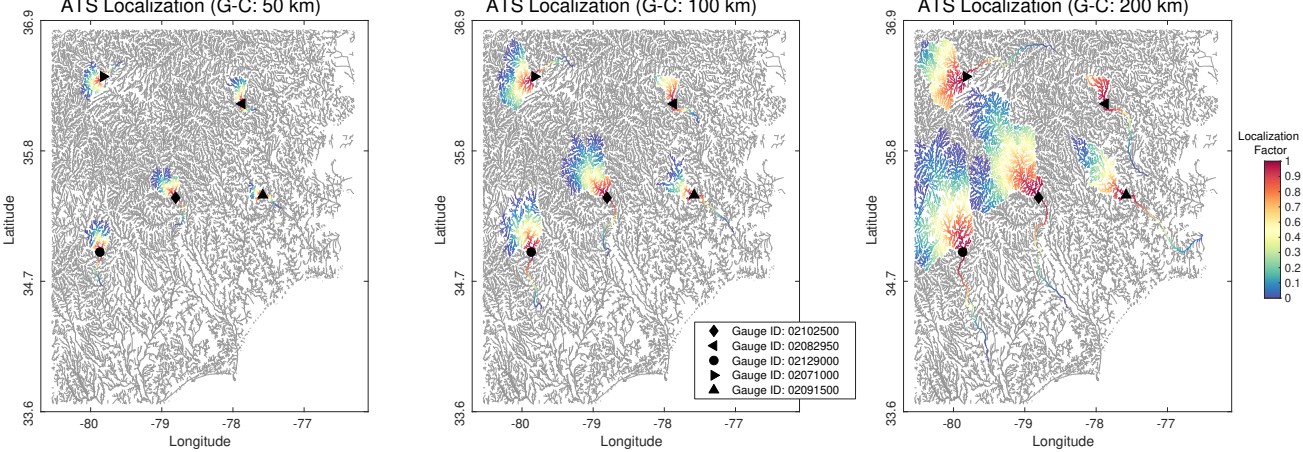

**Figure 4.** Illustration of the Along-The-Stream (ATS) localization strategy in the model domain using 3 different effective localization radii: 50 km (left panel), 100 km (middle panel) and 200 km (right panel). The resulting localization factor $\alpha$ is displayed for 5 different stream gauges. The correlation function used to compute $\alpha$ is based on the Gaspari-Cohn (G-C) $5^{th}$ order compactly supported Gaussian-like curve.

reaches upstream of the observation is significantly larger than the number of close reaches in the downstream direction. (ii) The total number of close reaches does not necessarily increase as $r$ increases. For instance, as can be seen in Fig. 4 the number of close reaches to gauge ID 0210500 ($\diamond$ marker) using $r = 50, 100$ and 200 km is 185, 646, and 1126 reaches, respectively. The same is not true for gauge ID 02082950 ($\triangleleft$ marker) because of the limited number of upstream reaches within the catchment. (iii) Observations in different catchments do not have common close reaches. Gauge IDs 02102500 and 02129000 for $r = 200$
310  km clearly demonstrate this feature.

The proposed localization method shares a lot of similarities with that of Emery et al. (E20, 2020b). The fundamental difference is that we are tackling sampling errors in the forecast error covariances with each assimilation cycle. Our sample covariances are computed using the evolving ensemble unlike E20 in which the authors use time-invariant covariances. Given that the hydrologic model used in E20 is linear, their system is technically an optimal interpolation with fixed error statistics.
Localization in this context is used to address structural errors of the covariances and to compensate for time-invariant covariances. Another important difference is that our ATS localization approach can ensure that the impact of the observation decreases as the distance from the observation, both upstream and downstream, increases. For example, the fifth-order polynomial function of Gaspari and Cohn (1999) can be used to find the localization coefficients or other functional forms can be used. In E20, on the other hand, all reaches that are close to the observation are assigned the same weight (i.e., $\alpha = 1$).

### 3.2.1 Tuning localization parameters

We conduct five DA experiments to study the sensitivity of the chosen localization radius on the accuracy of the streamflow estimates. The tested localization radii are: 50, 75, 100, 150 and 200 km. The performance of each experiment is assessed at four different locations inside the Florence domain (refer Fig. 1). Time-series of hourly forecast root-mean-squared-errors (RMSE)



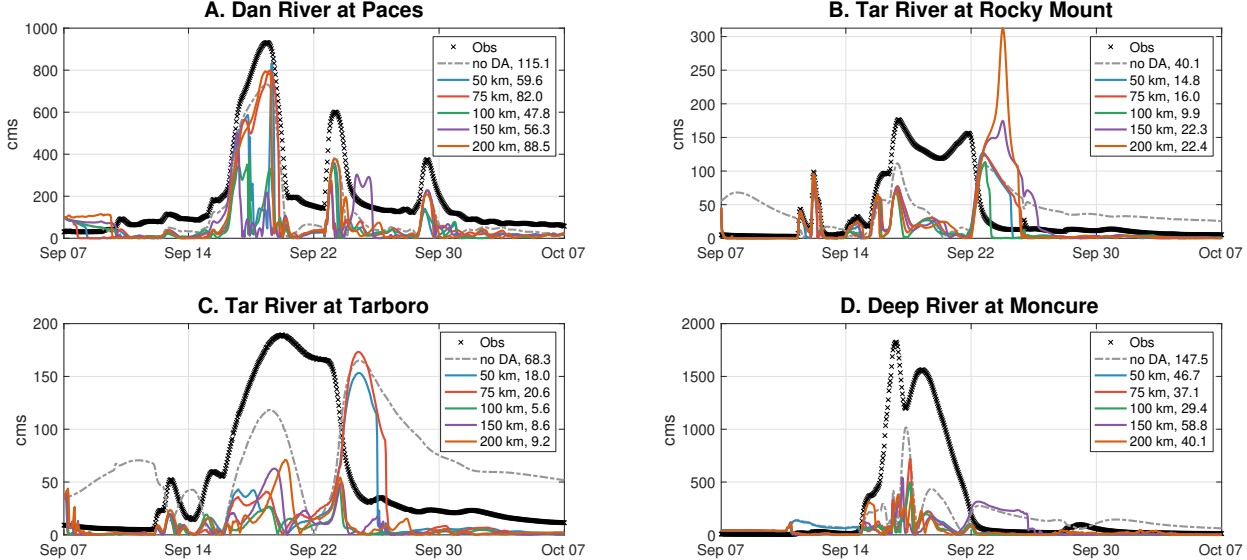

**Figure 5.** Time-series of hourly forecast RMSE of five DA experiments using ATS localization with Gaspari-Cohn function and radius $r = 50, 75, 00, 150, 200$ km . For each gauge, the observations are plotted by black "x"s. The other curves show the RMSE of each run. For example, the open loop (ensemble run with no DA) RMSE is given by the dashed gray curve. The time-averaged RMSE for each run is reported in the legend.

are displayed at each gauge in Fig. 5. As can be seen, all DA runs clearly outperform the open loop (i.e., no DA) especially

during the main event at around Sep. 17. Concerning the localization radius, it is shown that DA runs using $r = 50, 75$ km produce the least accurate streamflow estimates. At Tar River (Tarboro) for example, the RMSE from these two experiments is almost similar ($\sim 160$ cms) to that of the open loop on Sep. 25. This suggests that the Kalman update with such a small localization radius may be inadequate. Larger localization radii, $r = 150, 200$ km produce on average slightly better estimates. Such a performance, however, is inconsistent in space as can be seen at Tar River (Rocky Mount).

Overall, the best performance is obtained using $r = 100$ km. This was confirmed not only at the diagnosed gauges in Fig. 5, but also at the majority of the other available gauges in the domain (not shown). Our analysis suggests that larger radii generally give rise to spurious correlations and smaller ones limit the amount of useful information, both of which degrade the quality of the streamflow estimates.

The effect of the choice of the correlation function, used in the ATS localization scheme, is also investigated. We compare 3

different functions: Gaspari-Cohn, simple Boxcar (similar to E20) and a Ramped-boxcar. The resulting prior ensemble mean from each scenario is evaluated at all 107 streamflow gauges in the domain and the results for $r = 100$ km are summarized in the Taylor plot (Taylor, 2001) of Fig. 6. The diagram is useful to quantify the degree of correspondence between the gauge observations and the prior streamflow estimates in terms of 3 statistics: the Pearson correlation coefficient, centered root mean squared error, and the standard deviation. We note that boxcar and ramped-boxcar estimates produced erroneous

results at few (around 7) gauges and these results had to be removed from the diagram for visual purposes. Averaging over all

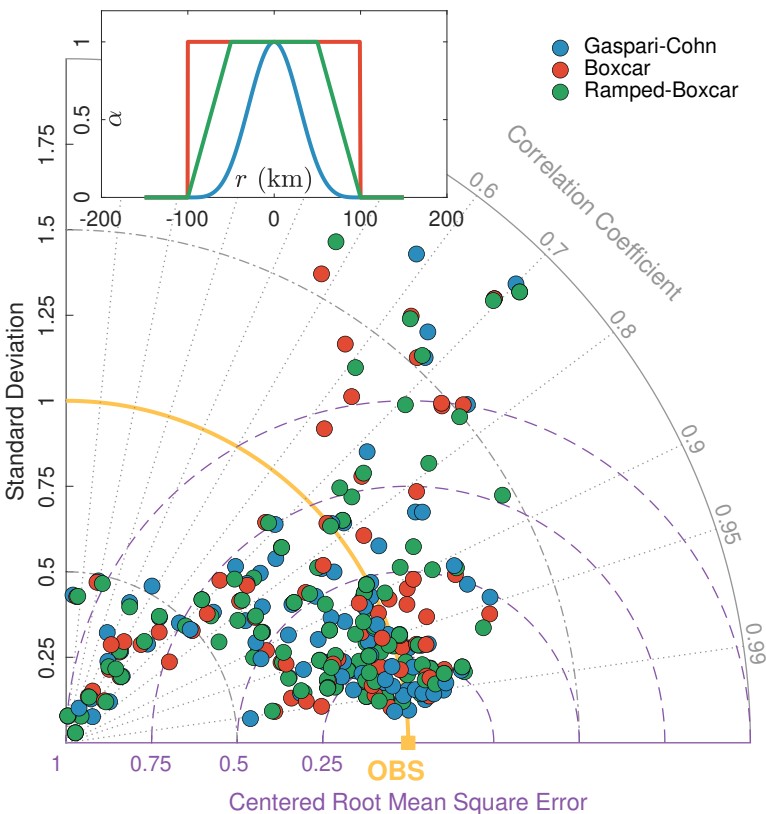

**Figure 6.** Taylor diagram for hourly prior streamflow estimates using ATS localization with 3 different correlation functions: Gaspari-Cohn, Boxcar and Ramped-Boxcar. The localization radius is set to 100 km. The shape of the functions are compared at the top of the plot. Ramped-Boxcar decays linearly to 0 starting at half-width (i.e., 50 km) distance from the observation. Comparisons to all gauges in the domain is performed, however, estimates with high errors and standard deviations, resulting from Boxcar and Ramped-Boxcar, are not shown for clarity.

gauges, the correlation coefficient resulting from Gaspari-Cohn, Boxcar and Ramped-Boxcar was found to be 0.83, 0.77 and 0.79, respectively. Gaspari-Cohn function further yielded the lowest values, on average, for the other 2 statistics of the Taylor diagram.

Boxcar and Ramped-Boxcar functions only outperform Gaspari-Cohn for small localization radii (e.g., $r = 50$ km), however, 345 the results produced with Gaspari-Cohn and $r = 100$ km have the best accuracy. This configuration, as a result, is selected and used in all other results shown in this study.

### 3.2.2 ATS Comparison to Euclidean Distance-based Localization

This section compares the proposed topologically based ATS localization to the regular Euclidean distance-based localization. Instead of searching for close streams on the river network as in the previous section, the regular approach looks for close by 350 reaches with a circle given a prespecified localization radius. Five different localization radii; namely $r = 1, 2, 5, 10, 20$ km are



**Table 2.** Comparison of ATS and regular (Reg) localization at Tar River and Deep River. The localization radius used in the ATS approach is 100 km. For the regular localization approach, 5 different radii are tested; namely 20, 10, 5, 2 and 1 km. The metrics used to compare the schemes are: prior and posterior RMSE, prior and posterior bias in addition to prior and posterior spread. The metrics (in cms) are all averaged over the entire simulation period.

| | | ATS (100 km) | Reg (20 km) | Reg (10 km) | Reg (5km) | Reg (2 km) | Reg (1 km) |
|---|---|---|---|---|---|---|---|
| Tar River at Tarboro (NWIS 02083500) | Prior RMSE | 5.579 | 18.541 | 8.860 | 33.459 | 41.607 | 34.323 |
| | Posterior RMSE | 4.930 | 17.819 | 6.748 | 25.106 | 33.664 | 26.411 |
| | Prior Bias | -1.130 | -11.648 | -1.706 | -20.242 | -18.091 | -11.068 |
| | Posterior Bias | -0.848 | -11.410 | -0.740 | -20.373 | -17.163 | -10.005 |
| | Prior Spread | 1.919 | 3.291 | 2.803 | 10.895 | 10.839 | 9.535 |
| | Posterior Spread | 1.551 | 3.004 | 2.271 | 6.283 | 6.425 | 5.170 |
| Deep River at Moncure (NWIS 02102000) | Prior RMSE | 29.440 | 48.328 | 69.206 | 102.992 | 130.085 | 146.354 |
| | Posterior RMSE | 22.749 | 36.163 | 38.009 | 38.169 | 38.159 | 63.991 |
| | Prior Bias | -6.498 | -14.880 | -21.001 | -38.676 | -58.086 | -75.678 |
| | Posterior Bias | -3.328 | -11.287 | -6.914 | -4.558 | -4.075 | -30.536 |
| | Prior Spread | 15.598 | 22.425 | 36.784 | 60.877 | 77.549 | 70.834 |
| | Posterior Spread | 11.867 | 16.812 | 19.645 | 21.434 | 22.038 | 25.595 |

tested. The resulting streamflow estimates are summarized and compared to the ATS (100 km) run, for two gauges, in Table 2. The two gauges, shown in Fig. 1, are selected such that the performance is assessed at relatively low (i.e, Tar River at Tarboro) and high (i.e., Deep River at Moncure) streamflow regimes.

Among the five experiments that use regular localization, the best performance is suggested using $r = 10$ km. As $r$ decreases below 10, the quality of the prior and posterior streamflow estimates diminishes. For instance, the time-averaged prior RMSE for $r = 10$ km and $r = 1$ km at Tar River is 8.86 cms and 34.323 cms, respectively. It is also noticeable that smaller localization radii yield large prior and posterior ensemble spread. This happens because the tiny localization radius tends to limit the impact of the data during the update and hence the shrinkage of the uncertainty around the ensemble mean gets restricted. For $r = 20$ km, the performance strongly degrades at Tar River. For example, the posterior bias is shown to grow from -0.74 cms using $r = 10$ km to -11.41 cms for $r = 20$ km. The reason for such a behavior is that with a radius of 20 km, streamflow gets falsely updated with information from nearby basins. Although these basins are physically close, however, they are governed by different flow regimes. The same is not true at Deep River and that is because the basin which Deep River belongs to is much larger and thus a localization radius of 20 km cannot contaminate the streamflow as we described at Tar River. Increasing the localization radius beyond 20 km yielded catastrophic updates for the streamflow and the subsurface bucket state. In fact, all DA experiments run with regular localization and $r > 20$ km failed at different stages (typically 2 weeks into the run).

Prior and posterior streamflow results obtained using ATS localization are significantly better than those with the regular localization. Unlike regular localization, using the proposed ATS approach we are able to increase the effective search radius



because the algorithm adheres to the physical aspects of the streamflow problem. Compared to the 10 km regular localization run, ATS produces at least 40% more accurate (in terms of RMSE) streamflow estimates. This is consistent for all 107 gauge

locations. Because the algorithm allows the use of large localization radii, ATS scheme further yields more certain estimates (smaller spread) than those that use regular localization.

## 3.3 Inflation

Variance underestimation in ensemble Kalman filters is a common issue that usually happens in the presence of large sampling errors and model biases (Furrer and Bengtsson, 2007). Sampling errors are the result of using a limited ensemble size. Model

biases are deficiencies in the model causing predictions to be far from the observations. Other sources of errors that might degrade the performance of the filter include non-Gaussianity (Anderson, 2010), systematic errors in the observational operator and representativity errors (Hodyss and Nichols, 2015). In practice, studies have shown that when model biases exist they tend to dominate other errors in the system (e.g., El Gharamti, 2018) and thus, treating model errors is often prioritized.

In this section, we consider three approaches to deal with the issue of variance underestimation: prior inflation (PR-inf),

posterior inflation (PO-inf) and combined prior and posterior inflation (PP-inf). In PR-inf the prior ensemble is inflated while in PO-inf the posterior ensemble is inflated. In PP-inf, before the update the prior ensemble is inflated and then the posterior ensemble is inflated after the update. In their recent study, El Gharamti et al. (2019) compared the three approaches in an atmospheric application. The authors argued that PR-inf is effective at mitigating model errors while PO-inf can only tackle sampling errors and other issues associated with the analysis such as non-Gaussianity. Combining both inflation schemes was

shown to produce the best results in application to atmospheric general circulation models.

The algorithm used to compute the inflation is adaptive in time, based on Bayes' theorem as in eq. (18), and results in spatially varying inflation fields.

$$p\left(\lambda|d^{f|a}\right) \approx p\left(d^{f|a}|\lambda\right) \cdot p(\lambda) \tag{18}$$

The algorithm assumes the inflation to be a random variable with an inverse-gamma prior distribution $p(\lambda)$. A Gaussian

likelihood function $p\left(d^{f|a}|\lambda\right)$ is constructed using forecast or analysis innovations, $d^{f|a}$, the observation error variance, and the variance of the ensemble. The posterior distribution of the inflation is obtained by taking the product of the likelihood and prior densities of $\lambda$ as shown in equation (18). To find the updated value of the inflation, $p\left(\lambda|d^{f|a}\right)$ is maximized and the resulting value is used as a prior for the next DA cycle. More details can be found in El Gharamti et al. (2019).

The hydrographs in figures 7 and 8 compare the performance of PR-inf, PO-inf, PP-inf with a no inflation (NO-inf) case at

395 two gauges along the Neuse River. At the upstream gauge (near Clayton, Fig. 7), the open loop shows a phase misalignment with the observations where the model floods almost a week after the main event on the ∼Sep. 15th. The hydrograph resulting from the NO-inf run is hugely biased as can be seen on Sep. 22nd. Because of the large discrepancies between the model estimates and the observation, the filter rejected almost 60% of the data[1]. PO-inf estimates are slightly better than those of

---

[1]Observation rejection (aka outlier threshold) in DART is applied when the distance between the ensemble mean and the observation is larger than 3 times the total spread. The total spread is computed as the square-root of the sum of the prior variance and the observation error variance.

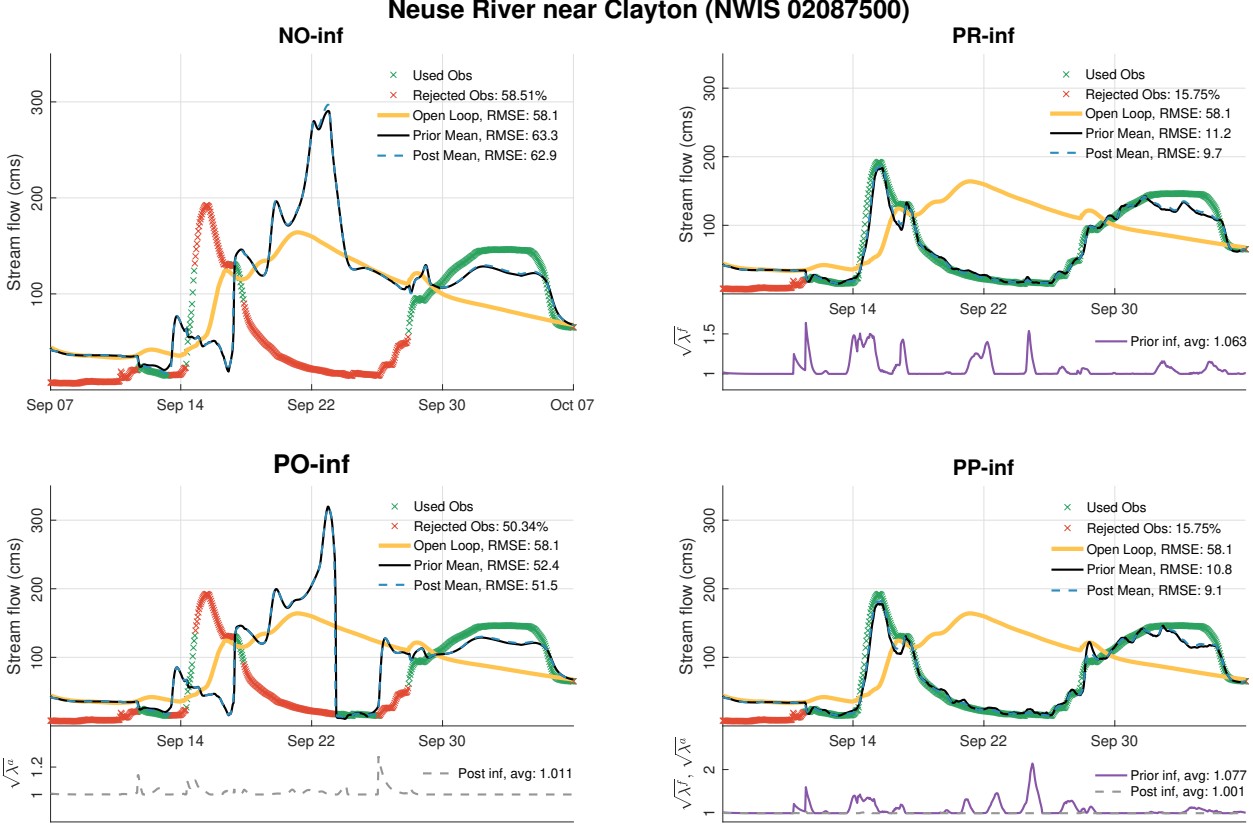

**Figure 7.** Time-series of prior and posterior ensemble means, at the upstream gauge "Neuse River near Clayton", resulting from 4 different DA runs: no inflation (top-left), prior inflation (top-right), posterior inflation (bottom-left) and prior and posterior inflation combined (bottom-right). The open loop hydrograph (in cms) is also shown. Assimilated and rejected observations are shown in green and red asterisks, respectively. The inflation mean time-series is plotted according the right y-axes. Time-averaged RMSE for each hydrograph is reported in the legend. The average values of prior, $\sqrt{\lambda^f}$, and posterior, $\sqrt{\lambda^a}$, inflation are also given in the legend.

the NO-inf run, however, almost half of the observations are still rejected. Using prior inflation (PR-inf) the majority of the
observations are assimilated producing high quality streamflow estimates. As can be seen, the large biases between Sep. 14[th]
and Sep. 22[nd] are completely removed. Whenever the model prediction starts to deviate from the observations' trajectory, the
adaptive inflation algorithm reacts immediately by restoring enough spread to bring the ensemble closer to the data during the
update. Once the model predictions become consistent with the observations, the inflation relaxes to smaller values. A value of
1 means no inflation is applied. The best fit to the observations is demonstrated by the PP-inf run. Its overall prior and posterior
averaged RMSE values are slightly better than those obtained using the PR-inf run.

At the downstream gauge (near Goldsboro as shown in Fig. 8), the discharge is almost 4 times larger than the upstream
gauge and the overall model fit to the data looks better. Towards the end of the flooding event (around Sep. 30[th]), PO-inf
better delineates the data compared to the NO-inf case. However, a false modeled flood wave appears during this time in

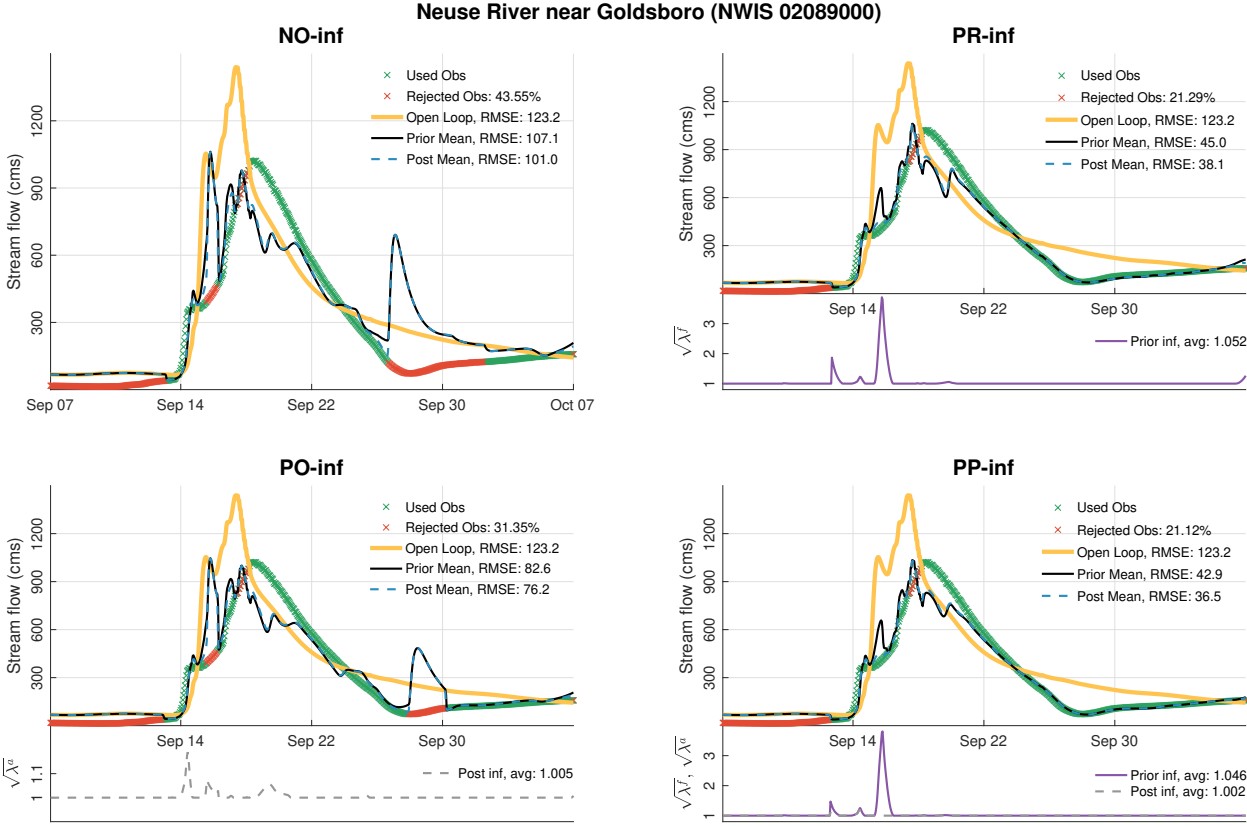

**Figure 8.** Similar to Fig.7 but for the downstream gauge "Neuse River near Goldsboro."

both simulations. Similar to Fig. 7, PR-inf run clearly outperforms the NO-inf and PO-inf runs yielding an average prior and

410 posterior RMSE of 45 and 38.1 cms, respectively. On average, the PP-inf prior and posterior estimates are $\sim 5\%$ more accurate than those of the PR-inf. Both PR-inf and PP-inf runs assimilate almost $80\%$ of the available hourly observations.

### 3.4 Choosing the best inflation

The results shown in figures 7 and 8 clearly demonstrate the usefulness of prior inflation at mitigating model biases. The benefits of using posterior inflation are only minimal. To illustrate how important prior inflation is, one could check out the

415 rising limb of the hydrograph at Neuse River near Goldsboro on Sep. 15[th]. With no inflation, the filter estimates are shown to overestimate the observed discharge and follow the trajectory of the open loop. Although $\sqrt{\lambda^a}$ is shown to increase to almost 1.2 in the PO-inf run, it is insufficient to bring the streamflow closer to the data. Assessing the PR-inf run, one could see that as the prior innovations begin to increase the adaptive scheme counteracts this by increasing $\sqrt{\lambda^f}$ to almost 4. As a result, the posterior mean is kept close to the data and consequently the prior estimates improved in the proceeding DA cycles.

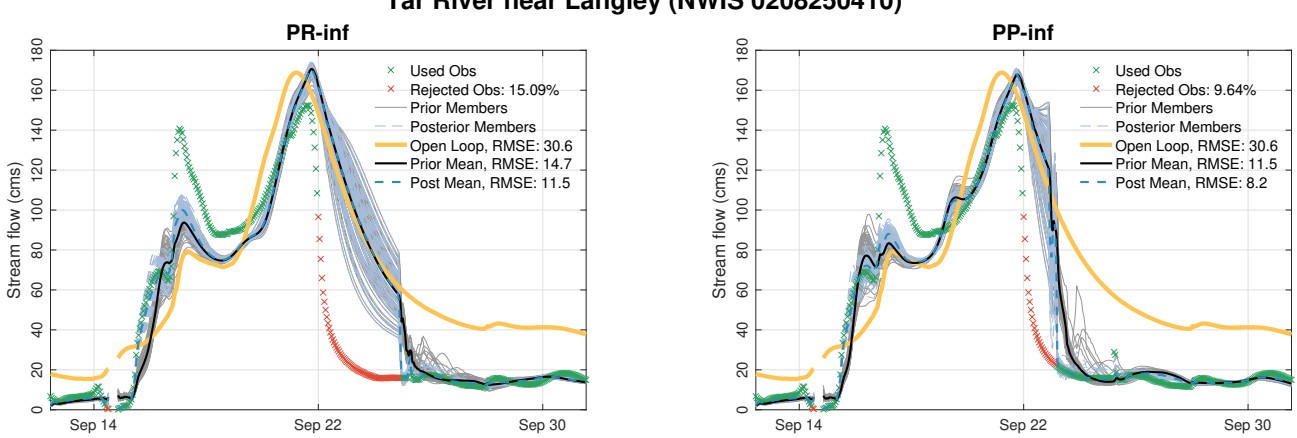

**Figure 9.** Similar to Fig.7 but for Tar River near Langley. Only the PR-inf (left-panel) and PP-inf (right-panel) results are shown. In addition to the ensemble means, the prior and posterior members are also plotted. Discontinuities in the hydrograph means that there are no observations to compare to.

Consistent with the findings of El Gharamti et al. (2019), adding posterior inflation on top of prior inflation further increased the accuracy. This suggests that posterior inflation might be resolving other regression issues such as sampling noise and non-Gaussianity. In fact, the gain from using posterior inflation on top of prior inflation is more pronounced at other gauges as shown in Fig. 9. As can be seen, PR-inf completely misses the falling limb of the hydrograph starting from Sep. 22[nd] to Sep. 25[th]. Prior and posterior means are shown to overlap with the open loop discharge. In the PP-inf run, on the other hand, the falling limbs of the simulated hydrographs are more similar to the data. The recession happens almost 2 days earlier and this in turn helps the filter reject less data (9.6% compared to PR-inf run's 15.1%) and produce higher quality estimates. The same behavior was observed at a few other gauge locations in the domain (not shown).

Computationally, combining both adaptive prior and posterior inflation schemes is more expensive than running each scheme alone. Our experiments suggest that the extra wall-clock time required to perform a full PP-inf run is around 20% of the total computing time required by PR-inf or PO-inf. In the current framework, the higher complexity is not found prohibitive especially when one takes into account the performance benefits that PP-inf provides. As a future study, it would be interesting to run other PP-inf cases with smaller ensemble size - to match the cost of the PR-inf run - and investigate the performance.

### 3.5 Inflation in space

The adaptive inflation varies spatially. With each cycle a different inflation factor is assigned to each value in the state vector. Using cross-correlations in the joint covariance, inflation is therefore computed not only for streamflow but also for the bucket portion of the state. Fig.10 maps the prior inflation for both streamflow and bucket obtained using PP-inf run. The displayed inflation field is an average over all fields obtained during the flooding period; i.e., between Sep. 12 and Sep. 18. Because of the localized update, the displayed inflation patterns generally follow the tree-like localization shapes (Fig. 4). Inflation values





tend to increase near the observation locations and decrease away from the gauges. This is why many reaches, especially in
the north east part of the domain, have no inflation (i.e., $\sqrt{\lambda^f} = 1$). Given the hourly assimilation of streamflow data, bucket
inflation values are relatively smaller than the streamflow ones. Streamflow inflation at more than 90% of the reaches do not
exceed the value of 2. Reaches with very large inflation values are located in densely observed areas, the inflation helps restore
ensemble spread after multiple, sequential state updates results in loss of spread.

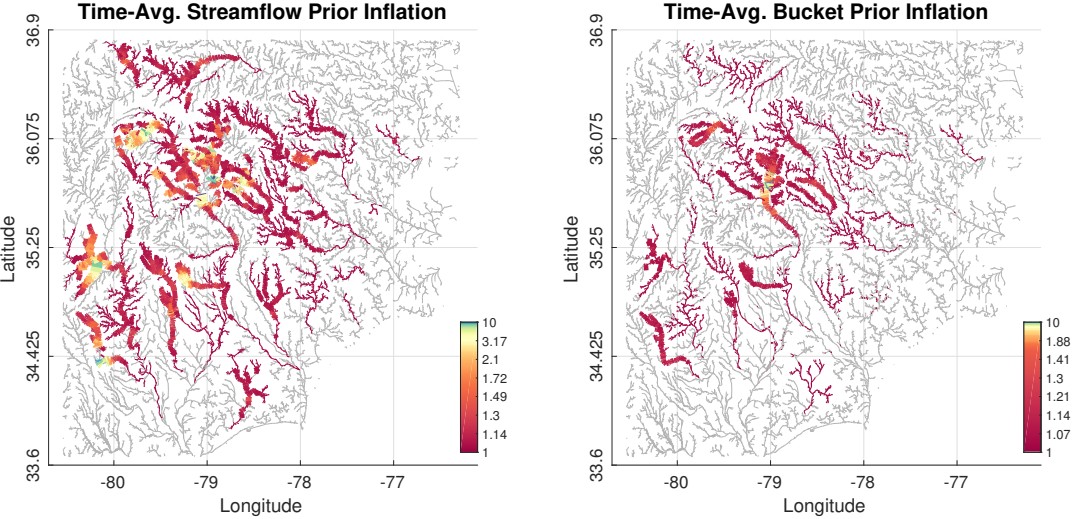

**Figure 10.** Time-averaged prior inflation for streamflow (left panel) and the bucket (right panel) resulting from PP-inf run. The inflation is
averaged over all estimates between Sep. 12 and Sep. 18. Color (log-scale) and line thickness both indicate the inflation value. Gray reaches
have inflation value of 1.

## 3.6 Overall Assessment

Prior to the the hurricane landfall on Sep. 14, streamflow estimates of the model appeared relatively good. The major differences
between observed and modeled streamflows resulted from the hurricane. The impact of DA prior to the hurricane is marginal.
To investigate this further, we show posterior streamflow maps on Sep. 13, 15 and 17 in Fig. 11 (top row panels). We also
show the difference between the posteriors and the open loop estimates (bottom row panels). Before flooding took place, the
highest flow was observed along Pee Dee, Cape Fear and Neuse Rivers as shown on Sep. 13. The difference between the DA
result and the open loop is confined to Cape Fear River and is equal to $\sim 200$ cms. Predicted streamflow on more than 70% of
the reaches in both runs is identical and hence the difference is shown to be 0 cms. The differences grow near Neuse River on
Sep. 15 to around 1000 cms. The posterior estimate of the streamflow in the rest of the domain on Sep. 15 is generally larger
than the open loop (mostly bluish in color). It is notable that streamflow in the domain increased by a factor of 7 before (i.e.,
maximum of 308 cms on Sep. 13) and after (i.e., maximum of 2170 cm on Sep. 15) landfall. On the 17th of Sep. the spatial
flow distribution changed considerably especially near the north-western side of Pee Dee River in which posterior streamflow




increased to nearly 7000 cms. Open loop streamflow estimates are surprisingly small in that area unlike the rest of the flooded domain.

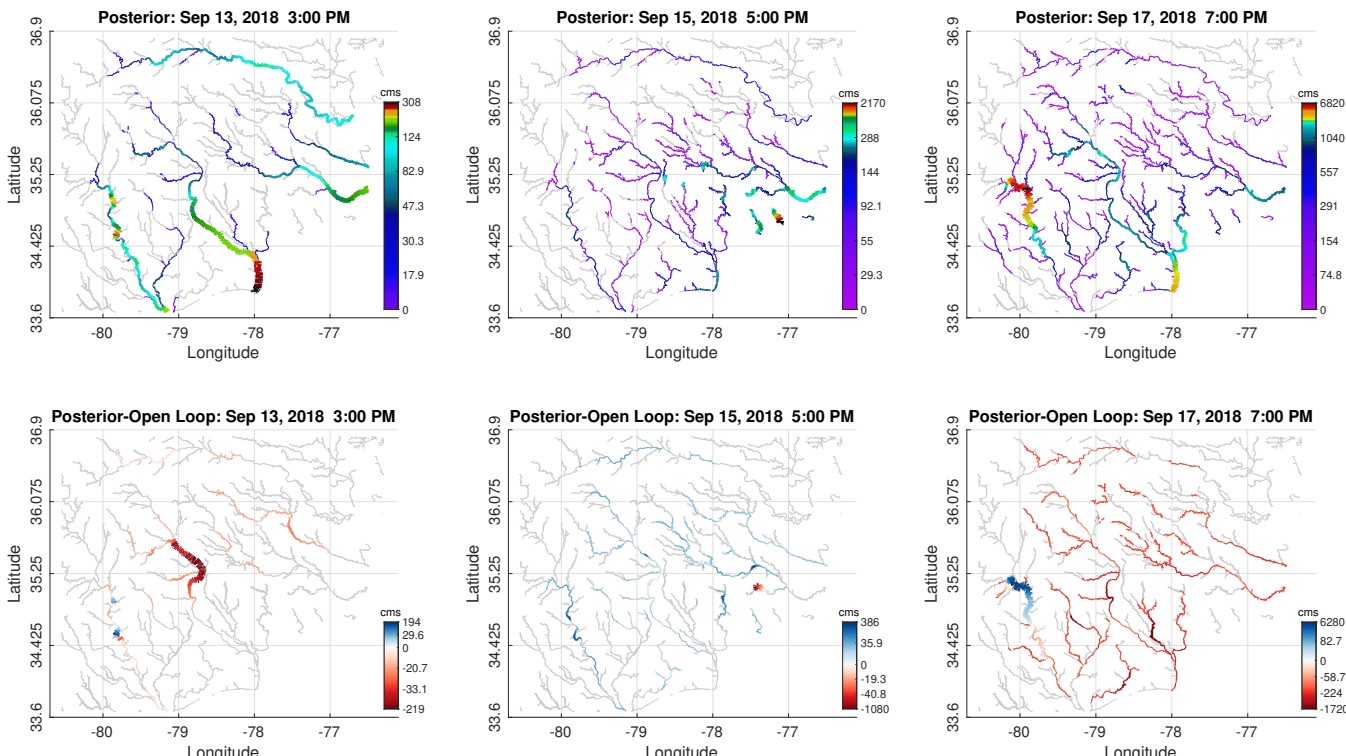

**Figure 11.** Top row panels: Posterior streamflow ensemble mean maps resulting from a PP-inf run on Sep. 13$^{th}$ at 3 PM (left), Sep. 15$^{th}$ at 5 PM (middle) and Sep. 17$^{th}$ at 7 PM (right). Bottom panels: Similar maps but for the difference between the Posterior means and the open loop estimates. Reaches with 0 cms flow are shown in gray color. Color (log-scale) and line thickness both indicate the magnitude value.

In order to understand the huge discrepancy between the posteriors and the open loop results, we study the streamflow evolution at Rocky River (just north of Pee Dee) in Fig. 12. On top of streamflow, we display the mean areal precipitation rates that are used to force the hydrologic model upstream of the gauged streamflow point. As can be seen, the open loop hydrograph severely underestimates the observed discharge on Sep. 17. While the observed discharge reaches 3000 cms, the open loop estimate does not surpass 100 cms. The reason for this huge bias is mainly attributed to the inaccurately specified rainfall rates which do not exceed 10 mm/hr during this time. Before this period, on Sep. 10, the forcings also falsely simulate heavy rainfall (around 40 mm/hr) prior to the hurricane's landfall. It is surprising how well the DA prior and posterior estimates are given these errors in the precipitation forcing. In fact, by looking at the RMSE values one finds that the prior and the posterior estimates are 56% and 90% more accurate than the open loop. Such a significant enhancement is obtained due to a massive inflation that gets applied to the prior streamflow ensemble. As shown in the right panel of Fig. 12, the inflation mean on Sep. 17 grows to 15. This growth was accompanied with a sizable increase in the inflation standard deviation. This further illustrates



how powerful the adaptive inflation algorithm in tackling large biases in the model. It is important to note that if the inflation
variance was fixed in time, then the inflation mean will not have the room to grow as much and hence the fit to the observed
discharge will not be as good. The posterior inflation mean values during the flood (not shown) were ranging between 1 and
2. In terms of ensemble spread, due to inflation the DA estimates are almost 2 orders of magnitude larger than the open loop
during the flood. The posterior ensemble spread is consistently smaller than the prior given the continuous hourly shrinkage
caused by the Kalman update.

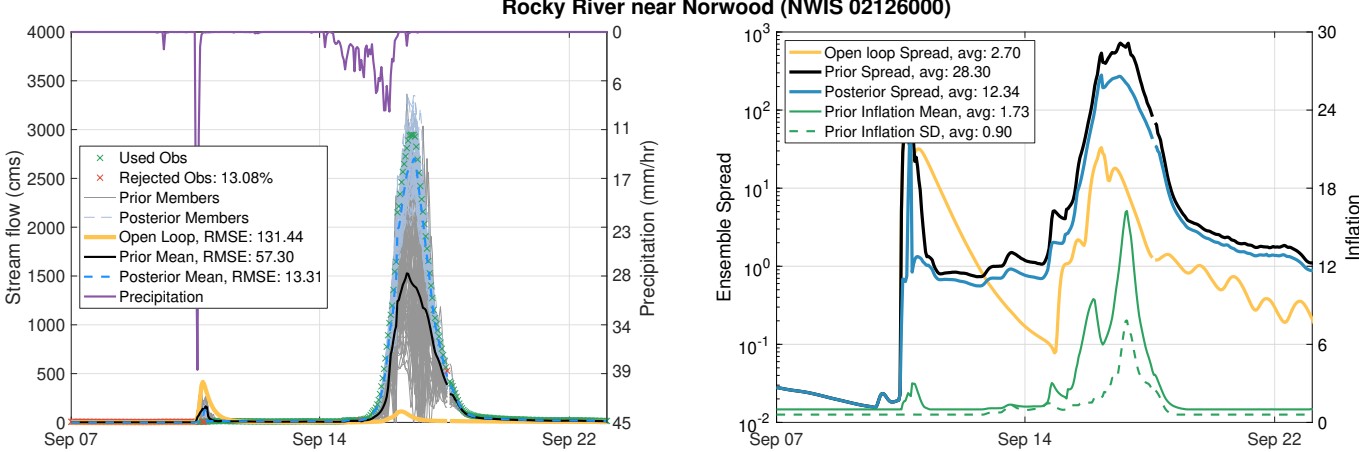

**Figure 12.** Left panel: Open loop, prior and posterior hydrographs obtained at the Rocky River near Norwood. Precipitation rates computed
at that gauge are plotted according the right y-axis. Right panel: Open loop, prior and posterior ensemble spread (in cms) time-series are
displayed in log-scale. Evolution of prior inflation mean and standard deviation (SD) is also shown. RMSEs, average ensemble spread and
average inflation mean and standard deviation values are reported in the legends.

The rank histogram is a useful statistical approach to visualize the behavior of the model and the priors along Pee Dee
River. The observed streamflow is binned with respect to the open loop and prior ensemble members at a single gauge near
Bennettsville, SC. The resulting probability bar diagrams are shown in Fig. 13. If the observation at all times falls within the
span of the ensemble members then one would expect to get a flat rank histogram. This is in fact exactly what we obtain for
the prior streamflow ensemble (bottom panel), making the observed discharge statistically indistinguishable from the ensemble
members. As for the open loop, the rank histogram suggests that the probability of the observation falling outside the open loop
ensemble is larger than 50%. The rank histogram for the open loop is heavily skewed to the right indicating that the observations
are most frequently much larger than the ensemble members, consistent with our previous analysis. The high probability in the
first bin of the histogram reflects the open loop's overestimation of the observed streamflow during the no-flood period.

To further assess the performance of the presented DA framework, we run an additional PP-inf experiment and instead
of assimilating all 107 gauges we withhold 3 gauges for validation. By withholding gauges, we can infer the impact of the
assimilation methods on ungauged points within the domain. The regime at the withheld gauges ranges between relatively low
flow at the Buffalo Creek, moderate flow at Lumber River and high flow at Cape Fear River. Linear regression is performed

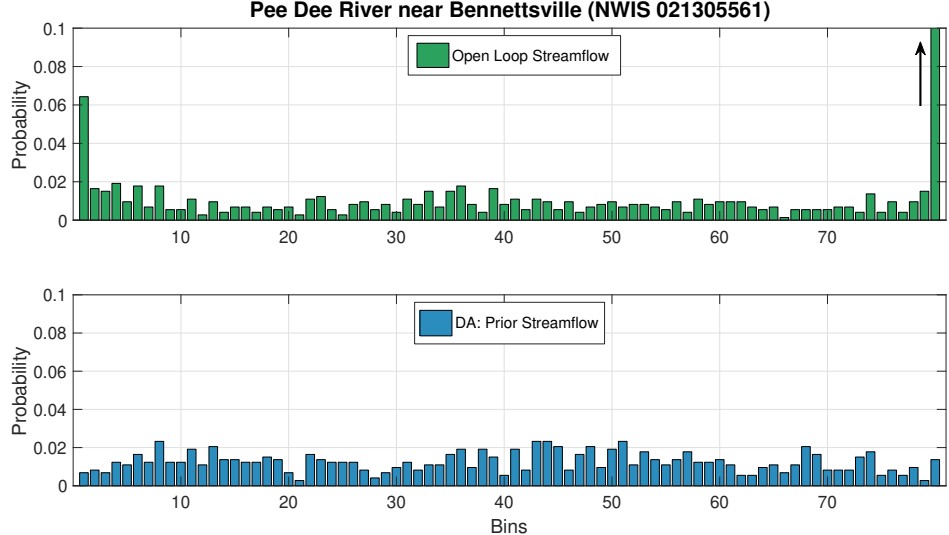

**Figure 13.** Rank histograms for the open loop and the prior streamflow obtained at Pee Dee River near Bennettsville. The histograms have been normalized to show probability instead of the observation count. The arrow in last bin of the top panel indicate that the probability is large ($\sim 0.5$).

to validate the streamflow estimates obtained using the open loop, PP-inf (assimilate all 107 gauges), and PP-inf-w (withhold 3 gauges) at the withheld gauges. The resulting analysis is shown in Fig. 14. The PP-inf run is presented as the best case

scenario. We check if PP-inf-w can outperform the open loop and how well it approximates PP-inf. Compared to the open loop, the performance of PP-inf-w at the withheld gauges is considerably more accurate. At Lumber River, for instance, the open loop shows a strong overestimation of the observed discharge strongly improved in both assimilation runs. For all 3 gauges, the estimates from PP-inf-w are able to reasonably mimic those of the PP-inf. Overall, PP-inf-w yields better RMSE and more desirable $R^2$ (coefficient of determination) values than the open loop. This result indicates that the streamflow at

unobserved locations is significantly improved by the assimilation.

## 4   Summary and Discussion

NOAA's National Water Model configuration of the WRF-Hydro framework is coupled to the Data Assimilation Research Testbed (DART) to improve ensemble streamflow forecasts under extreme rainfall conditions during Hurricane Florence in Sep. 2018. Hourly streamflow and bucket head states are simulated using a channel+bucket submodel of the NWM. These

states are then updated through data assimilation (DA) using streamflow observations collected from 107 USGS gauges. The system uses 80 ensemble members, incorporating multiphysics uncertainty (each ensemble member assumes different channel model parameters) and time-varying uncertainty in the forcing fluxes to the channel and the bucket models.





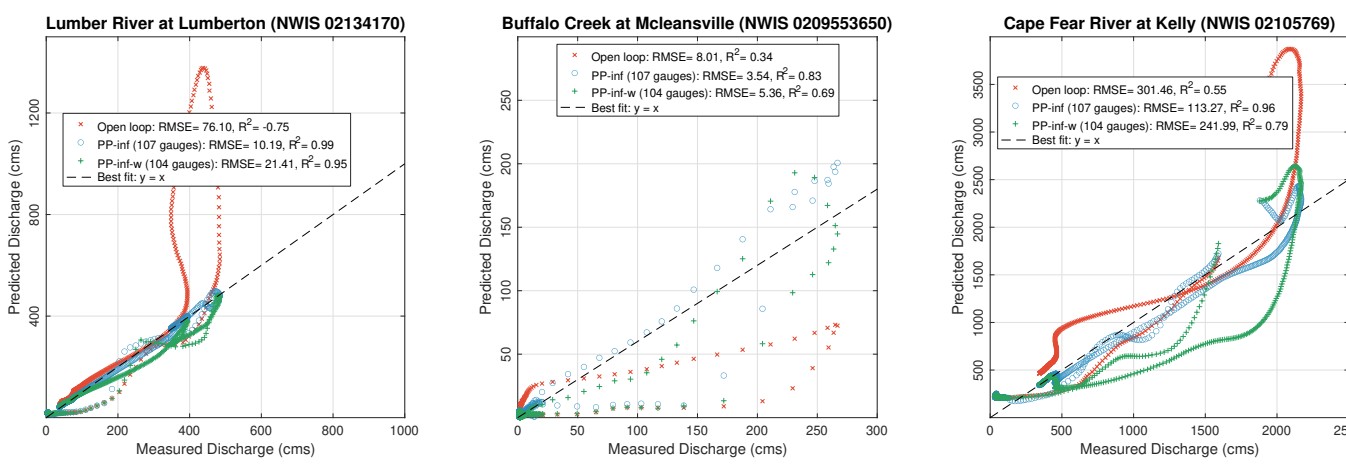

**Figure 14.** Cross plots of the streamflow at 3 withheld gauges. The results are shown for the open loop, the PP-inf (where all 107 gauges are assimilated) and PP-inf-w (where only 104 gauges are assimilated). The best fit line is denoted by a black dashed line. Average RMSE value and the coefficient of determination ($R^2$) are computed and reported in the legends.

This study presents two main contributions within a generalized ensemble DA framework for hydrologic systems, particularly those defined on irregular grids such as a stream network. First, a topologically based "along-the-stream" (ATS)
localization is shown to improve information propagation during the model state. Localizing the impact of the update mitigates sampling errors due to undersampling as well as other analysis errors. Moreover, ATS localization specifically eliminates error-covariances between unconnected streams. The algorithm requires tuning of a localization radius and we do not attempt to diagnose a physical basis for estimating the optimal radius *a priori* (as such a discussion should probably include estimation of temporal error covariances not considered in this study). However, ATS localization was found to produce results signifi-
cantly better than the regular Euclidean distance-based approach. The improved results stem in part from a larger localization radius under the ATS approach, indicating more effective propagation of the observations in the update along the stream than through Euclidean space. While the ATS approach does not further the cause of "predictions in ungauged basins", it indicates that further research into novel localization strategies for streamflow DA may bear additional fruit. On this point, we note that the impact of the ATS localization strategy on the results of this study relative to the impact of adaptive inflation and bias
correction is remarkably larger than would be expected in application to atmospheric DA.

The second major contribution of our study is to demonstrate utility of spatially and temporally varying adaptive inflation (El Gharamti, 2018) in hydrologic applications, particularly to help control model bias. Prior and posterior adaptive inflation is shown to mitigate model biases and sampling errors, respectively. Results during major flooding events illustrate that severe model biases can be effectively reduced using adaptive prior inflation. Because the method is spatially varying, different
degrees of bias in different parts of the stream network can be efficiently tackled. Posterior inflation was not found as effective as prior inflation, however, combining both inflation schemes yielded the highest streamflow accuracy. Overall, inflation plays





an indispensable bias correction role, without which, the quality of the ensemble streamflow prediction would best be described as poor.

To validate the results of the presented DA system, a variety of diagnostics are presented. Hydrographs at different locations in the domain were investigated. Prior and posterior streamflow estimates were compared to the open loop result. The largest streamflow improvements were found along Pee Dee River in South Carolina after landfall, during which the observed streamflow was strongly underestimated by the open loop. Improvements due to assimilation were also demonstrated using a rank histograms at a gauge along Pee Dee River. Streamflow and inflation spatial maps were also analyzed. It was found that streamflow inflation values are larger than those of the bucket state, given that streamflow is directly observed. The overall changes to the bucket state after DA were minimal. To test the impact of DA at non-observed locations, 3 gauges were withheld from the assimilation and the resulting prior estimates were verified against the data. Linear regression tests revealed that observations at nearby gauges are able to improve the streamflow at the location of the withheld gauges, eventually reducing the systematic biases of the open loop.

The most challenging aspect of the hydrologic DA is the problem of model biases or errors. These biases are usually associated with inaccurate boundary conditions (e.g. precipitation), uncertain parameters (e.g. channel roughness and slope) or model physics deficiencies. This study has shown that adaptive inflation can prove effective at handling biases in the data assimilation. Apart from inflation, a handful of other techniques can be performed to mitigating bias issues. Jointly estimating highly uncertain model parameters alongside the state is an approach commonly found in hydrology (e.g. Vrugt et al., 2006; Gharamti et al., 2015; Abbaszadeh et al., 2018; Ziliani et al., 2019). Updating parameters often increase the complexity of the DA framework (nonlinearity often increases in state-parameters estimation problems) and the computational cost may become prohibitive, especially for spatially varying parameters. Yet, such an approach may yield improvements to the analyses of this study. The multiphysics approach considered here aims to incorporate uncertainty in the fixed boundary condition (geometry, roughness parameters) into the ensemble in order to better model the background error covariance. A combined multiphysics and joint parameter estimation approach might also be pursued. Uncertainty of updated parameters tends to dissipate in time and may be more appropriate for certain kinds of conceptual model parameters instead of those considered in our multiphysics approach. Further up the model chain, not considered in this study, running WRF-Hydro with a land surface would allow for updating of soil moisture and surface head states. Instead of treating deterministic fluxes with parameterized noise, introducing these prognostic variables would provide the ability to adjust the fluxes coming to the channel. Many studies have tried this and remarked on the problematic updating of soil moisture from streamflow due to the highly nonlinear relationship between the states, particularly for flood forecasting applications (Rakovec et al., 2015). While expanding the prognostic states of the model may potentially improve aspects of the flood prediction problem, such as overland and subsurface fluxes to the channel routing configuration, it is possible that shifting the boundary conditions up the model chain may result in a similar bias issue with more degrees of freedom in the state vector. Coupled atmospheric and hydrologic DA would be a further step towards updating the prognostic states causing hydrologic errors in the state vector. These are ideas to be pursued in future studies.

An essential DA ingredient this study did not cover is Gaussian anamorphosis (Simon and Bertino, 2009; Gharamti et al., 2017). Streamflow, being strictly non-negative, is a nonGaussian variable. Since the Kalman update is linear and assumes





Gaussian statistics, it becomes more appropriate to transform streamflow to a Gaussian space where the update is performed and then it can be pulled back to the physical space. This is well known in hydrological applications (e.g. Clark et al., 2008). Such a transform guarantees that the updated streamflow does not consist of any unphysical (i.e., negative) values. The transformation
is often conducted using empirical functions or analytical ones such as the natural logarithm. This will be investigated in a follow-up study.

Finally, one-hour ahead (prior) forecasts of flooding event were the focus of this study. Future research will study the impact of DA in the whole forecast time window up to 18 hours in the short-range forecasts, and expand the DA application to medium- and long-range forecasts including additional hydrologic components and observations. The functionality of the ATS
localization and inflation may change in different forecasting modes. For instance, longer localization radii could be found more desirable in a long range forecast.

*Code availability.* The data assimilation code used in this study is openly available as part of the DART repository (master branch) on GitHub; `https://github.com/NCAR/DART/tree/master/models/wrf_hydro`. The model code is also freely available and can be accessed through GitHub; `https://github.com/NCAR/wrf_hydro_nwm_public`.

*Author contributions.* MEG developed the localization and the inflation algorithms, ran the DA experiments and wrote more than 60% of the manuscript. JLM developed the channel+bucket submodel, the python framework that configures and runs wrf_hydro with DART, and wrote Sections 1 and 2. SJN contributed to the introduction Section and the discussion of the results. TJH provided significant help with the DART code. AR helped retrieve the observations and the precipitation data. BKJ built an OSSE framework that provided helpful insights to the real DA system.

*Competing interests.* The authors declare that they have no significant competing financial, professional, or personal interests that might have influenced the performance or presentation of the work described in this manuscript.

*Acknowledgements.* The authors would like to thank Jeffrey Anderson and David Gochis for fruitful discussions. We are grateful to Nancy Collins for her assistance with coding the recursive search algorithm of the localization scheme. We also would like to acknowledge high-performance computing support from Cheyenne (doi:10.5065/D6RX99HX) provided by NCAR's Computational and Information Systems
Laboratory, sponsored by the National Science Foundation. Any opinions, findings, and conclusions or recommendations expressed in this publication are those of the authors and do not necessarily reflect the views of the National Science Foundation.



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
