# Peer review of "Ensemble Streamflow Data Assimilation using WRF-Hydro and DART: Novel Localization and Inflation Techniques Applied to Hurricane Florence Flooding"

_Hydrology and Earth System Sciences, 2020_

## Author Comment (AC1)

We would like to thank the reviewer for their positive feedback and suggestions. All of the reviewer's comments have been addressed below. Please find the response in blue.

- Please provide a summary table for the input dataset used in this study and its specifications.

| Observation Data | |
|---|---|
| *Variables* | - Streamflow/discharge (m³/s)
- Station IDs
- Time |
| *Source* | United States Geological Survey: https://waterdata.usgs.gov/nwis |
| *Number of Stations* | 107 |
| *Time period* | September 1 - October 15, 2018 |
| *Frequency* | Gauge dependent |
| *Formats* | Binary data gathered using dataRetrieval R package; converted to DART-style hourly observation sequence files |

We constructed the above table in response to the reviewer's request. This exercise helped us improve and clarify some of the language used to describe the observations in Section 2.7 and also throughout the paper. However, we have chosen to not include the above table as part of the manuscript as the data are relatively simple to understand and we believe they are sufficiently summarized in the text.

- There are so many acronyms used in this paper; please try to avoid this.

The reviewer is right, we may have used a lot of acronyms in this paper but most of these refer to model or software names (e.g., WRF-Hydro, DART, Noah-MP), datasets (e.g., NHDPlus, RAP, HRRR), centers (e.g., NOAA, NWIS, USGS) or techniques and metrics (e.g., DA, M-C, RMSE). All of these acronyms are widely used in the literature and avoiding them makes the text quite long and cumbersome. Only few new acronyms were introduced in this study: ATS, PR-inf, PO-inf and PP-inf. We appreciate the reviewer's comment and we'll try to avoid excessive use of acronyms in future work.

- Why did the authors choose the Muskingum-Cunge Streamflow model to route the flows? For example, this method has limitations in backwater effects, flood plains storage and interaction of channel slope in hydrograph. Several dams and reservoirs are present in this region, and therefore, the backwater effect might affect the flow routing.

The choice of the Muskingum-Cunge streamflow model was made as part of the design of NOAA's National Water Model. In this paper, we investigate applying data assimilation to this particular model. A streamflow routing model with backwater effects and linkages to floodplains and lakes would indeed provide a more realistic simulation, but would be difficult to run at the scale of the NWM. The increased physical realism may provide additional challenges for instantaneous data assimilation which are not considered in our application. We focus on the streamflow routing model used in the NWM to be relevant to that model.

- If there is the streamflow's baseflow underestimation, how the NWM model solved this issue? Please clearly explain in the methodology section.

We believe that the answer to this question is present in the text already:

"The NWM employs a groundwater bucket model as a simple aquifer representation to mitigate this baseflow problem." and "The bucket scheme is simple and highly conceptual. For this reason, calibration of its parameters is critical for reasonable model simulations."

Which is to say that the model calibrates baseflow performance through the parameters of the bucket model. There is no additional mechanism in the model for adjusting baseflow. We hope this answers the reviewer's question.

- How is the ensemble of 80 members selected in this work? The author has written that this number is achieved based on the computational demand and statistical performance. I would expect at least a figure to justify this optimization.

We thank the reviewer for bringing this up. This is a great suggestion. In the revised manuscript, we added a new figure (also below) where we compare the performance (in terms of prediction skill score) and the computational demand (as function of both time and number of CPUs). We run additional experiments using 40 and 160 members and compare the results to the 80-member experiment we already had.

[Figure]

We argue that using 80 members produces estimates that are almost as good as those obtained using 160 members. Furthermore, the 80-member ensemble run cuts down on the computational demand (of the 160-member run) quite significantly. Lastly, one could further reduce the computational effort massively using 40 members however this sacrifices the accuracy of the streamflow. In short, we find an ensemble size of 80 an optimal way to balance the performance and the computational demand. A similar analysis was added to Appendix A of the revised manuscript.

- Why are multipliers sampled using a uniform distribution? If another distribution is used instead of the uniform distribution, how will it affect the final results?

The main reason for using uniform distribution to sample the channel parameters is because uniform pdfs offer an easy procedure to sample bounded quantities. The channel parameters are bounded from above and below and in this case the six of them are all positive. In addition, because the parameters are geometric ones, they are sampled under some physical constraints (for example, top width cannot be smaller than bottom width). Lastly, the channel parameters are unknown quantities and hence using bounded uninformative priors (such as uniform pdf) is a reasonable choice. Other distributions such as Gaussians can be utilized but one needs to make sure the sampled values do not fall outside the predefined physical bounds and more importantly the draws have to be positive. By doing so, we would be effectively utilizing biased and truncated Gaussians (not pure Gaussian pdfs). Other forms such as beta, lognormal or exponential distributions could be used but were not explored in this study. We don't believe the choice of the sampling distribution will have major effects on the streamflow results. In future studies, we could look into this in more detail. Thank you!

---

## Author Comment (AC2)

We would like to thank the reviewer for their valuable time and efforts in reading the manuscript and providing helpful review of the work. We have carefully considered all the suggestions and modified the manuscript accordingly. We believe that these comments have greatly improved the quality of the manuscript. Detailed response to all reviewers' comments and concerns are given below.

Main comments:

1. Novelty: the abstract lists the two methodological innovations mentioned above. From the literature review in the introduction it is however not entirely clear to what extent these are innovations. For example, the text (lines 60-67) does not specify whether space-time adaptive covariance inflation has been applied to flood forecasting, except that "the impact of inflation on streamflow predictions is not fully understood", without providing a reference. Similarly, regarding ATS covariance localization, the introduction (lines 78-82) does not mention whether this technique is proposed here for the first time or if it has been applied in other studies. So, my suggestion is to make the exact contributions of the paper more explicit.

We have followed the reviewer's suggestions and re-wrote part of the text in the Introduction. Regarding inflation, we added the following (Lines 66-68): "**This is the first study of its kind where spatially and temporally adaptive inflation is applied to streamflow forecasting. This study further explores the use of prior versus posterior inflation and investigates the effect of each scheme on the performance of the flood prediction ensemble framework.**"

As for ATS localization, the text has been modified as follows: "**We investigate updating distributed hydrologic states and propose a new topologically-based localization strategy for stream networks. The method is called along-the-stream (ATS) covariance localization and it confines state updating to directly connected (defined below) hydrological states.**"

2. Title of the paper suggests that the main innovation of the paper is data assimilation during an extreme rainfall/flooding situation (hurricane), whereas abstract/intro focuses on methodological innovations as the main contribution (see point 1). Please clarify/make it consistent.

We do see the value of indicating the main innovations of the paper in the title, as you suggest. The original title of the paper is:

"Ensemble Streamflow Data Assimilation using WRF-Hydro and DART: Hurricane Florence Flooding"

Originally, we felt it was useful to indicate the application of the data assimilation in the title of the paper and did not mean to suggest flooding application as an innovation. Here is the new title:

"**Ensemble Streamflow Data Assimilation using WRF-Hydro and DART: Novel Localization and Inflation Techniques Applied to Hurricane Florence Flooding**"

Now we hope the title conveys both the innovation and the application context. While covariance localization is a general problem in ensemble data assimilation, the use of adaptive inflation is meant specifically to deal with potentially large model errors in the flooding context that we investigate. We added brief note to that effect:

Line 13 (abstract): "We demonstrate that ATS localization provides improved information propagation during the model update. Adaptive prior inflation is used to tackle errors in the prior, including large model biases **which often occur in flooding situations**."

Thank you for this suggestion.

3. Methodology: key novel parts of the methodology are not described in sufficient detail, specifically the ATS covariance localization strategy in section 3.2 and the inflation method in section 3.3 (see more details below).

Below please find a detailed response to all concerns raised about the Methodology.

4. Results: the provided results and figures do a good job of illustrating the benefits of the proposed methods, so no major comments in this respect. I do however have some comments on providing additional results, see below.

We performed additional runs as requested. Further details can be found below.

Detailed comments (more or less in chronological order):

- line 83: typo in "hydrological"

Done.

- line 164: "The coefficients in equation 2 can be found in the literature", please provide a reference

The earlier Ponce and Yevjevich (1978) reference covers this but we added a more accessible Ponce and Lugo (2001) reference as well that shows these equations. We have removed reference to "the literature" and clarified that it is just a derivation:

"**The coefficients in equation (2) can expressed as combinations of the Courant and Reynold's numbers (e.g., Ponce and Lugo, 2001), respectively**"

- line 221: include units for these values

The units for E and G are given on line 217 (right before equation (11)). The unit for z_max in mm has been added.

- line 224: "We did not investigate the effect of ensemble size on the results within this study". Ok but it would still be helpful to address whether a larger ensemble size would change the results and conclusions of the paper. E.g. since the methods aim at fixing sampling errors due to small ensemble size (among other things), does their benefit decrease with larger ensemble size?

Based on the reviewer's comment (and Reviewer #1 concern), we have run additional experiments in the revised manuscript using 40 and 160 ensemble members. We compared the results to the 80-member ensemble we've already been using. As we describe in Appendix A (c.f. Fig. A1) and in our response to the first reviewer, 80 members is selected as an optimal solution to balance the statistical performance and the computational demand.

The reviewer also brings another good point regarding sampling errors (and other issues) and the effect of changing the ensemble size on the functionality of the inflation and localization techniques. Increasing the ensemble size should help fix most of the ensemble filtering issues. On the other hand, decreasing the ensemble size would further deteriorate the results due to large sampling errors. For example, we tuned ATS localization for the 40-member ensemble run and found that 50 km gives the best estimates (recall that the 80-member runs used 100 km effective localization radius). The 160 ensemble produced the best results when using 100 km, similar to the 80-member ensemble. This is because ATS localization is tackling not only spurious correlations but also other issues in the analysis step of the EnKF such as regression errors, nonGaussianity (also nonlinearity), etc.

[Figure]

Apart from sampling errors, increasing the ensemble size does not solve the issue of model biases. That's why we use space and time adaptive inflation and we argue that the technique can act as a bias correction scheme. In our experiments, we found that increasing the ensemble size can actually reduce the need for inflation. The adaptive scheme still uses inflation but the magnitude of the inflation values decrease as the ensemble size increases. As can be seen in the figure below, the inflation values (averaged in space) for Ne=160 are slightly less than those obtained using Ne=80, however, inflation is still needed especially during the flooding period where severe biases arise. For the 40-member run, the adaptive scheme uses inflation values that are larger than the other 80 and 160-member runs in order to account for biases and additional sampling errors.

All in all and to address the reviewer's comment, varying the ensemble size does not change the conclusion of the presented study. Because the inflation scheme is adaptive, it will use inflation that is best suited for the selected ensemble size. ATS localization, just like regular localization, needs to be tuned based on the choice of the ensemble. Even for large ensemble sizes, localization may still be needed to tackle other issues that one usually ignores.

- line 227: priors on stream channel parameters are missing from fig. 2

This is a good suggestion that illustrates the "multiphysics" streamflow ensemble. We have added the parameters to the figure and in the caption, we noted:

**"The depicted time-invariant *a priori* error distribution of channel parameters provides a "multiphysics" streamflow ensemble."**

- line 269: not clear what is meant by "bucket distributions"

To avoid confusion, we have omitted the word "distributions." Both streamflow and bucket variables are included in the DART state.

- eq.14: subscript k is not defined

Subscript k, time index, is defined right after equation (2). Both model and data assimilation equations make use of the same indexing.

- I was wondering whether the USGS streamflow data (rating curve based) are still accurate during a hurricane. Reading section 2.7 it seems the answer is "no", since the paper uses revised streamflow data. On line 253 it is stated that using original vs revised streamflow data had significant impact on the results, yet only results with revised data are presented. Do the methods proposed here still work when using the original non-revised data? And if not, why not and what is needed? This should then be addressed in the discussion as an open problem for DA under realistic hurricane conditions.

We agree that the discussion can be expanded to address these questions. We actually have a paragraph in the source (manuscript) file that we commented out previously for the sake of brevity. We have made the following changes based on the request of the reviewer:

"The USGS streamflow observations used by the NWM are provided along with its output in near-real-time on NOMADS. The streamflow observations in these files, which correspond very closely to the values assimilated by the NWM, are always "provisional" because they are near-real-time and they are subject to revision until they have been thoroughly assessed. For this study, we collected NWM observation files as well as revised values from the USGS's NWIS many months after the time period of this study. As expected, there were significant revisions to the streamflow values in the months following Hurricane Florence. **These revisions are for multiple reasons, not the least of which is that existing rating curves do not typically extrapolate well to extreme and out-of-bank flows. We note that the difference between these observation sets had a significant impact on our results and that the provisional data proved more challenging for the assimilation methodology in this paper. It is extremely important to study the differences between such provisional and approved data in order to bridge the gap between the methods offered in this paper and real-time data assimilation applications. Ultimately, one would want to assimilate provisional data and evaluate against revised data. There are multiple issues to consider in this regard including observation gaps, uncertainty, and quality measures. In our study, we chose to use the revised observations to evaluate the performance of our methodological innovations. This study could be extended to simulate real-time streamflow assimilation.** "

- line 260: "observation error plays a somewhat secondary role in the quality of the assimilation, ". I guess that is after revision of the original streamflow data!?

The reviewer is right in the sense that if we were to use the real-time provisional data, then a lot of tuning effort needs to be done in order to optimize the observation error variance. With the revised observations, the observation error variance is fixed throughout the flooding period (i.e., 20% of the flow) and we rely solely on the adaptive space-time inflation to adjust the spread of the ensemble.

- line 276: define 'observation increments'

The observation increments are now defined. We follow the least squares approach of Anderson (2003) and partition the assimilation problem into 2 steps: The first updates the ensemble members of the observed variables and the second step regresses these updates to state space. This has been clarified and a reference has been added.

- line 279: "alpha is computed...", this sentence remains very cryptic and unclear at this point. Suggest to refer to section 3.2 where it is explained in more detail.

A reference to Section 3.2 has been added.

- eq. 17: in this paper, is inflation applied to forecast, analysis, or both? Edit: ok later in the paper this becomes clear, but good to briefly mention here as well.

We have added a sentence in line 301 indicating that we perform both prior and posterior inflation in our experiments. A reference to the inflation experiments' section is also added.

- section 3.2 (localization): this section is a key contribution of the paper and needs to be better explained. The section describes the approach in words and illustrates what it looks like in figure 4 (nice figure), but it doesn't actually show how to implement it. Please include relevant mathematical expressions so that the methodology is reproducible.

The localization coefficient $\alpha$ is calculated for each streamflow gauge and the close reaches given a predefined correlation function. It depends on 2 parameters: (1) the distance between the observation and the state variable and (2) the localization radius. The mathematical expression for $\alpha$ has now been added for the 3 different correlation functions (i.e., Gaspari Cohn, Boxcar, Ramped Boxcar) in Table A1 of the revised manuscript.

-line 303: ATS localization assumes flow of information only travels downstream not upstream. Why is this a reasonable assumption and why is this assumption needed?

This is a good question. The premise as stated is not fully accurate and deserves some clarification in addition to the discussion of the assumption. The text in Section 3.2 has been modified accordingly.

"**ATS localization highlights some key features: (i) Upstream from each observation, information flows up the network, including through the bifurcations. Downstream from each observation, we assume that the flow of information only travels downstream with the observed flow. As such, we obtain tree-like shapes where the number of close reaches upstream (tree canopy) of the observation is significantly larger than the number of close reaches in the downstream direction (tree trunk). Not allowing information to "round the bend" or bifurcate back upstream below the gage, we choose to only update flows which contribute to the observation (upstream) and to which the observation contributes (downstream). This choice was made to be distinct from Euclidean distance-based localization and out of caution, given a modestly sized ensemble, that observations near the confluence of major tributaries might have undue influence on large flows with potentially low (true) error correlations. Allowing upstream bifurcations below the gage could be a reasonable approach as well, pending choice of ensemble size and understanding of correlated errors at major tributaries.**"

- line 331: why don't you show these results to corroborate your conclusion that r=100km works best? It may further be helpful to plot forecast performance as a function of localization radius for all gauges to corroborate the statement on line 332-333 that forecasts deteriorate for r<100 and r>100.

That's a good suggestion by the reviewer. We added another panel to Figure 5 where we summarize the forecast performance from all the gauges in the domain as a function of localization distance using boxplots. We also added 2-3 sentences to Section 3.2.1 to discuss the new panel. As the reviewer mentioned, this plot further supports our claim that 100 km yields the best streamflow estimates.

- section 3.2.2: this is a nice section that reports promising results of ATS vs regular localization (table 2)

We thank the reviewer for their complement. The results in this Section are indeed promising and we hope that this gets the attention of the Hydrology community.

- section 3.3 (inflation): similar to section 3.2, the inflation method should be better explained. Only Eq. 18 is now given, which shows the standard prior times likelihood formula (but note that the posterior on the left is proportional to, not approximately equal to, prior times likelihood). I suggest providing more details on the likelihood function: what does it look like, what are the underlying assumptions, why are the assumptions valid. Also, the difference between prior and posterior inflation should be better explained.

Following the reviewer's suggestions, we have now added the inflation likelihood function in equation (19) of the revised manuscript. The likelihood is Gaussian with first and second moments defined using observation-space diagnostics following Deroziers et al. (2005). The mean of this function is zero assuming uncorrelated forecast and observation errors. We argue that this is a valid assumption in most earth systems and we provide a solution for the relatively rare case when these quantities are correlated. Furthermore, we show the difference in the algorithm between prior and posterior inflation. In essence, instead of dealing with prior ensemble statistics the analysis ensemble is used. As such, the variance of the analysis innovations (i.e., the second moment of the Gaussian inflation likelihood) is computed as: $var(da) = o2-ya2$ unlike the prior inflation case where $var(df)= o2+yf2$. $o2$ is the observation error variance, $yf2$ is the prior ensemble variance and $ya2$ is the analysis ensemble variance. We also provide a reference (El Gharamti et al., 2019) for further details on the inflation algorithm.

- line 392: "To find the updated value of the inflation, p(λ|d) is maximized and the resulting value is used as a prior for the next DA cycle". But after maximization one (optimal) value is obtained, how can this single value be used as a prior distribution in the next DA cycle? It sounds like this approach does not keep track of the posterior of the inflation factor? Please clarify.

This is a good observation by the reviewer. The text has been revised to clarify this. After maximizing the posterior inflation density, the resulting value is used as the mode of the prior density in the next data assimilation cycle. We also keep track of the inflation standard deviation. As such, both the updated value of the inflation and its standard deviation are used to compute the parameters (i.e., shape and scale) of the inverse-gamma prior density. Further details are again referenced in the first author's previous work.

-line 546: "running WRF-Hydro with a land surface..." missing 'model'?

Done.

---

## Author Response (AR2)

We thank the reviewer for their comments and suggestions. Please find our answers below.

**Technical comments:**

1. Regarding the ATS methodology: if I understand it correctly, the distances used (r and xi in newly added Table A1) are the shortest linear distance between a gauge and a reach, not the distance along the river network? And so, the ATS approach can be understood as a 'standard' Euclidean localization approach but with reaches that do not meet certain topological constraints removed? Adding a short additional clarification on this point would be helpful.

The distance is actually computed along the river network. For example, if there are 3 reaches separating the observation gauge and the link we need to update, the distance (xi) is the sum of lengths of these 3 reaches. The data which contains the length of the river reaches within CONUS is part of USGS's National Hydrography Dataset (NHDPlus). This is now clarified in the caption of table A1.

2. Eq. 18: replace 'is approximately equal to' by 'is proportional to' symbol

Done.

3. There are missing citations in the text (appearing as '?' in the pdf, e.g. on lines 278, 321, and several other locations)

The reviewer is right. These showed up in the marked-up version of the manuscript due to a typesetting issue. The non-marked version didn't have that problem. In any case, thanks for bringing this up. We have now fixed all in-text citations in this final form.

4. Links are provided to source code for the hydrological model and data assimilation toolkit on GitHub. That's great. However, these are 'general' software code repos. In the spirit of open science, it would be helpful to also make available a GitHub repo with the specific implementation for the case study used in this paper, including the new ATS methodology which may not yet be available (?) in the existing repos.

All configuration, setup, running and diagnostic scripts used in this work are available in the provided DART repository. The proposed ATS localization methodology is also provided in this repository. Here is a list of some important codes and scripts:

- ATS localization: `${dart_path}/model_mod.f90`

- Setup an experiment: `${dart_path}/python/experiment/setup_filter_experiment.py`

- Run an experiment: `${dart_path}/hydro_dart_py/hydrodartpy/core/run_filter_experiment.py`

- Run diagnostics: `${dart_path}/matlab/run_HydroDARTdiags.m`